# ICQuant: Index Coding enables Low-bit LLM Quantization

**Xinlin Li[†], Osama Hanna[◇], Christina Fragouli[†], and Suhas Diggavi[†]**
[†]University of California, Los Angeles
[◇]Meta, GenAI
Email:{xinlinli, ohanna, christina.fragouli, suhasdiggavi}@ucla.edu

## Abstract

The rapid deployment of Large Language Models (LLMs) highlights the need for efficient low-bit post-training quantization (PTQ), due to their high memory costs. A key challenge in weight quantization is the presence of outliers, which inflate quantization ranges and lead to large errors. While a number of outlier suppression techniques have been proposed, they either: fail to effectively shrink the quantization range, or incur (relatively) high bit overhead. In this paper, we present ICQuant, a novel framework that leverages outlier statistics to design an efficient index coding scheme for outlier-aware weight-only quantization. Compared to existing outlier suppression techniques requiring $\approx$ 1 bit overhead to halve the quantization range, ICQuant requires only $\approx$ 0.3 bits; a significant saving in extreme compression regimes (e.g., 2-3 bits per weight). ICQuant can be used on top of any existing quantizers to eliminate outliers, improving the quantization quality. Using just 2.3 bits per weight and simple scalar quantizers, ICQuant improves the zero-shot accuracy of the 2-bit Llama3-70B model by up to 130% and 150% relative to QTIP (Tseng et al., 2024b) and QuIP# (Tseng et al., 2024a); and it achieves comparable performance to the best-known fine-tuned quantizer (Malinovskii et al., 2024) without fine-tuning. [1]

## 1 Introduction

Despite their success, the accessibility of Large Language Models (LLMs) is curbed by their significant memory and computational power requirements. Weight quantization, a technique that represents model weights using lower precision, is promising to alleviate these demands while preserving the model's performance. Quantization not only enables a reduced memory footprint, but also accelerates inference time as memory fetch latency is reduced. As a result, quantized models benefit from reduced energy consumption and lower running costs. Quantization opens the road for the deployment of LLMs over resource-constrained devices (for instance, smartphones that have memory and battery constraints), better positions LLMs to serve real-time applications (such as speech recognition and real-time translation that require fast inference times), and overall lowers the deployment cost, leading to more sustainable computing and broader usage.

Weight-only post-training quantization (PTQ) has emerged as a popular compression method (Frantar et al., 2022; Lin et al., 2024; Zhu et al., 2024). It avoids the heavy computational cost of retraining associated with quantization-aware training (QAT) and enables the on-demand quantization at a required precision. Yet, a major obstacle in achieving low-bit ($\leq$ 4 bits) weight quantization without significant performance deterioration is the presence of weights with exceptionally high magnitudes that reside in the tails of the weight distribution, known as outliers. Outliers significantly expand the required quantization range, making it hard to cover with a small number of bits without detrimental quantization errors.

Given its importance, a number of techniques have been proposed to mitigate the challenge of outliers. For instance, a common technique is to clip the outliers to fixed thresholds.

---

[1]Code available at: https://github.com/Avery-xl/ICQuant

Recent work, such as BitDistiller (Du et al., 2024) and OmniQuant (Shao et al., 2023), advanced this concept by optimizing the clipping range dynamically. However, the substantial quantization errors associated with outliers limit its practical application. Another prevalent approach is weight grouping (Frantar et al., 2022; Shao et al., 2023; Lin et al., 2024). Weight quantization typically occurs at per-(output) channel, and grouping further divides weights into small continuous blocks (e.g., with size 128, 64) to leverage the reduced local ranges. Our statistical analysis (in Section 2) reveals that the effectiveness of this method is limited, as outliers frequently persist within individual groups. Furthermore, the storage requirements for group-specific quantization parameters, such as scales, zero-points, and look-up tables, make this approach impractical for non-uniform and vector quantization schemes.

More recent approaches include SqueezeLLM (Kim et al., 2023) and SpQR (Dettmers et al., 2024), which maintain outliers (and/or important weights) in full precision (FP16). While effective, this method also incurs significant storage overhead due to storing the full precision outliers and their indices. QuIP (Chee et al., 2023) proposed an innovative incoherence processing technique, which employs random rotation matrices on both sides of weight matrices to suppress outliers. However, additional matrix operations introduce notable computational overhead during inference. Moreover, when many layers's weights exhibit independently and identically distributed Gaussian behavior (as observed in Dettmers et al. (2023)), such rotations often yield small improvement, as discussed in Appendix H.2.

In this paper, we introduce ICQuant, a novel quantization framework that enables us to deal separately with outliers with minimal storage overhead (approximately 0.3 bits per weight). In particular, ICQuant explicitly partitions the weights into outliers and inliers, keeps track of the positions of outliers using an optimized indexing scheme, and uses a different quantization codebook with reduced range for the outlier and inlier weights. A key insight on why our approach works well, is the empirical observation that, within each row of the weight matrices, the positions (indices) of outliers tend to follow a uniform distribution, and thus can be efficiently encoded. We verify this property across multiple models (including the Llama2, 3, 4, and Qwen2.5 families) that have a wide range of scales. We also note that, even if this property were not to hold in other/future models, a one-time random permutation can enforce uniformity without affecting model output and inference speed, and thus ICQuant would still work well.

Our contributions include:

- We introduce ICQuant, a novel outlier-aware LLM quantization framework that leverages statistical properties of outliers for effective weight quantization. An attractive feature of ICQuant is that it can be universally applied on top of any quantization scheme.
- We conduct extensive experiments showing that ICQuant can significantly improve the quantization quality, in 2-4 bits regimes, even of simple scalar quantizers, achieving comparable results to state-of-the-art schemes that rely on computationally intensive vector quantization and expensive fine-tuning procedures.
- We analytically derive an upper bound (see Lemma 1) on the bits required by our index coding scheme, which closely aligns with our experimental results, and amounts to $\approx 0.3$ bits per weight.
- Our work reveals that the outlier weight spatial distribution is well approximated by a uniform distribution, which may be of independent interest.

The rest of the paper is organized as follows. Section 2 presents the statistical properties of outliers that motivate our scheme; Section 3 introduces our design of ICQuant; Section 4 provides our experimental evaluation; Section 5 discusses additional related work, and Section 6 concludes the paper.

## 2 Statistics of Outliers

Although various techniques exist (as described in Section 1) to reduce the weight quantization range, the statistical properties of weight outliers, such as their distribution and impact

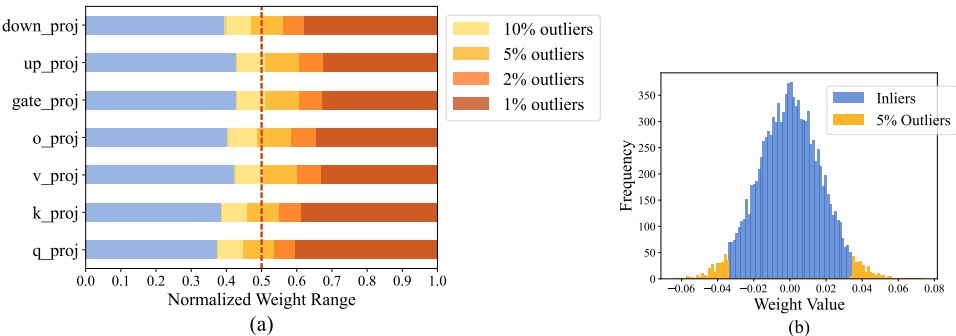

Figure 1: [Llama2-7B] (a) the normalized range taken by different amounts of outliers (starting from 1%), where the values of each type of layer are averaged over the whole model; (b) histogram of a row of weights.

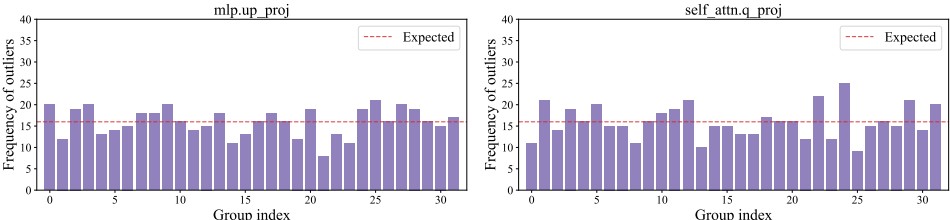

Figure 2: [Llama2-7B] The frequency of outliers in each group of 256 consecutive weights.

on quantization efficiency, remain poorly understood. This section examines these statistical characteristics in detail.

In this paper, we define outliers as the small percentage of weights with high magnitude (i.e., those in the tails of the weight distribution). Specifically, for a weight matrix $W \in \mathbb{R}^{d_{out} \times d_{in}}$, outliers are identified as the top $\gamma$ weights (e.g., $\gamma = 5\%$) with the highest absolute value in each output channel (i.e., each row $w \in \mathbb{R}^{d_{in}}$).

*1). 5% outliers take approximately the 50% range of all weights.*
Previous studies have focused primarily on a narrow subset of outliers ($< 1\%$), and we extend this analysis to examine the relationship between the proportion of outliers and the quantization range. Figure 1 (a) plots the normalized weight range taken by outliers across layers in Llama2-7B model (more detailed analysis provided in Appendix B), which demonstrates that the top 5% of weight outliers account for approximately 50% of the total value range $r$ (where $r = \max(w) - \min(w)$). Figure (b) provides a closer view, showing the distribution of (a row of) weights in Llama2-7B, where 5% of outliers are highlighted.

This observation reveals the significant inefficiency in low-bit quantization. For example, considering the uniform scalar quantization, using $n$ bits yields a resolution of $\frac{r}{2^n}$. If the range is halved, then $n - 1$ bits would suffice to reach an equivalent resolution $\frac{1}{2} \frac{r}{2^{n-1}} = \frac{r}{2^n}$. Our finding, therefore, indicates that one bit of quantization capacity is effectively allocated to represent just 5% of the weights. This creates notable inefficiency, particularly when $n$ is small (e.g., 2 or 3 bits are allowed in total).

| | q_proj | k_proj | v_proj | o_proj | up_proj | gate_proj | down_proj |
|---|---|---|---|---|---|---|---|
| Llama2 - 7B | 3.38% | 3.39% | 3.14% | 62.15% | 3.10% | 3.12% | 2.37% |
| Llama3 - 8B | 3.01% | 3.13% | 2.95% | 95.25% | 3.10% | 3.08% | 2.11% |

Table 1: The rejection rate (i.e., the percentage of weight channels where outlier indices do not follow a uniform distribution) according to the Chi-Square test with 0.05 significance.

*2). Outliers are spread out uniformly.*
More importantly, we found that the outlier indices are uniformly distributed across each output channel (i.e., each row of the weight matrix). In Figure 2, we show the frequency of outliers within each group of 256 consecutive weight elements. We validated this distribution using the Chi-Square test (Pearson, 1900), with results for Llama2-7B shown in Table 1 (see more results in Appendix C.1). The analysis reveals that around 97% of the weight outliers' positions follow a uniform distribution, except for out_projection layers in self-attention blocks. This uniform distribution pattern explains why simply using small group sizes for quantization (explained in Section 1) provides limited benefit in addressing the outlier issue. In the next section, we present an efficient coding scheme for storing outliers by leveraging this structural characteristic. Notably, we empirically observed that the deviations in out_projection layers have minimal impact on the overhead of our proposed coding scheme.

**Observation.** The uniform distribution of outlier positions likely stems from the transformer's Gaussian-like initialization (Glorot & Bengio, 2010) and over-parameterization (Kaplan et al., 2020) of large models. Yet we note that, even when this uniform structure does not naturally emerge, we can enforce it by randomly permuting the input channels of each linear layer in advance - a process that preserves both model output and architecture, as we discuss in Appendix C.2.

## 3  Methodology

In this section, we present a novel quantization framework ICQuant for low-bit ($\leq$ 4 bits) weight-only quantization of LLMs. Our approach preserves the effective quantization range through strategic separation of outliers and efficient index storage. The proposed method is compatible with various quantization schemes, including uniform/non-uniform scalar and vector quantizations.

We use $\mathcal{Q}(\cdot)$ to denote a quantization scheme (or quantizer), which is associated with a set of quantization parameters (such as scales, zeros, and lookup tables). For simplicity, we refer to all these quantization parameters as a codebook. We define the quantization size as the average number of bits used to store each quantized weight.

### 3.1  Outlier-Grouping Strategy

Building upon our first observation from Section 2, we propose separately quantizing the top 5% outliers and inlier values, with each utilizing approximately half of the quantization range for the models we quantize in Section 4. Given an output channel (i.e., a row) of weights $w \in \mathbb{R}^{d_{in}}$, we partition them into two groups $\{w_o, w_i\}$, where $w_o$ collects outliers and $w_i$ collects inlier values. These are quantized into $\{\mathcal{Q}_1(w_o,), \mathcal{Q}_2(w_i)\}$ using two independent quantizers $\mathcal{Q}_1(\cdot)$ and $\mathcal{Q}_2(\cdot)$ and the same number of bits. Our analysis (in Section 2) shows that 5% of outliers take approximately 50% of the weight range, namely

$$\text{range}(w_o) \approx \text{range}(w_i) \approx \frac{1}{2}\text{range}(w).$$

Figure 3 (a) demonstrates an example using rounding-to-nearest (RTN) uniform quantization, where we compare vanilla-RTN against our proposed approach. Notably, since the ranges of both $w_o$ and $w_i$ are halved, we employ 2-bit quantization in our method versus 3-bit quantization in vanilla-RTN. Figure 3 (c) visualizes the experimental results on a $w$ from Llama2-7B model, which shows that our INT2 quantization achieves comparable resolution to INT3 vanilla-RTN.

It is worth noting that traditional grouping requires $\frac{d_{in}}{g}$ codebooks, where $g$ is the size of each group. The storage overhead for one additional codebook in our case is negligible, which makes it suitable for non-uniform quantization or vector quantization schemes that typically use large codebooks.

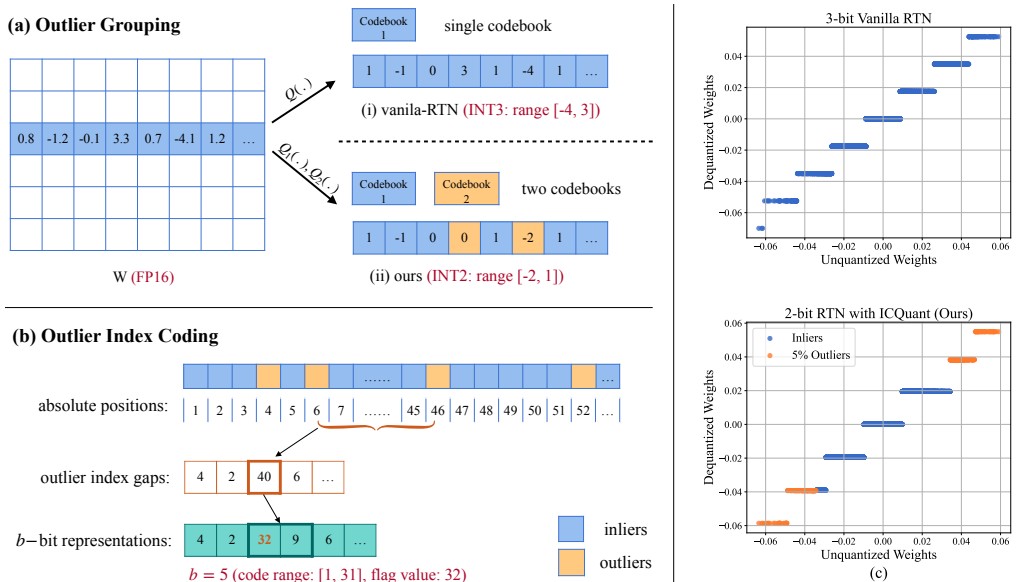

Figure 3: (a) Comparison of vanilla-RTN and ICQuant in 2-bit uniform quantization. (b) An example of our outlier index coding scheme. (c) Quantization results on a row of weights from the Llama2-7B model.

## 3.2 Efficient Outlier Index Coding

A critical challenge is how to efficiently store the outlier position information. Traditional methods such as binary flags or direct index storage are prohibitively expensive. Specifically, attaching a binary flag to signal outliers requires 1 extra bit per weight. Although storing outlier indexes needs much fewer entries, each entry requires at least 16 bits due to the large dimensionality of LLMs (e.g., $d_{in}$ reaching 50K in Llama3-405B model). This approach also results in roughly 0.8 extra bits per weight for 5% outliers for the Llama3-405B model.

Leveraging the uniform distribution pattern of outlier positions in $w$, as discussed in Section 2, we propose storing the index gaps between adjacent outliers rather than their absolute indices. This significantly reduces storage requirements. Let $\gamma$ denote the outlier ratio, $b$ denote the number of bits used to represent each index (gap), and $B$ represent the effective average index storage cost per weight (total number of bits to store outlier indices divided by the row size). Given weights $w \in \mathbb{R}^{d_{in}}$, let $\{i\}_{k=1}^{p}$ be the absolute indices of the outliers, where $p = \lfloor \gamma d_{in} \rfloor$. Now taking a simple example, if $\gamma = 5\%$ and the distance between each pair of consecutive outliers is up to 32, i.e, $i_{k+1} - i_k \in \{1, \ldots, 32\}, \forall k = 1, \ldots, p-1$ - then $b = 5$ bits (supporting the range $[1, 32]$) is sufficient to store each index gap, reducing the storage overhead to merely $B = \gamma b = 0.25$ extra bits per weight.

Although outliers are uniformly spread out, we need to address the practical randomness that the inter-outlier gaps vary. Consider the supporting range $[1, 2^b]$ of $b$-bit representations, gaps not larger than $2^b$ can be well covered. The challenge arises when the inter-outlier gaps exceed the $2^b$-unit threshold. To address this issue, we designate the value $2^b$ as a flag to indicate large gaps requiring index count accumulation, as illustrated in Figure 3 (b). As a result, we have $[1, 2^b - 1]$ left to represent the value of gaps. While this accommodation could potentially introduce additional storage overhead, we provide a tight upper bound on this cost in Lemma 1, with proof given in Appendix A.1.

**Lemma 1** *For a weight $w \in \mathbb{R}^{d_{in}}$ with $\gamma d_{in}$ outliers, if the weight values are randomly permuted (with uniform outlier positions) and $b$ bits are used to encode each index gap, the total storage overhead $B$ (defined as the number of bit per weight to store the outlier positions) satisfies*

$$\mathbb{E}_{uniform}(B) \leq \gamma b (1 + \frac{1}{e^{\gamma(2^b - 1)} - 1}).$$

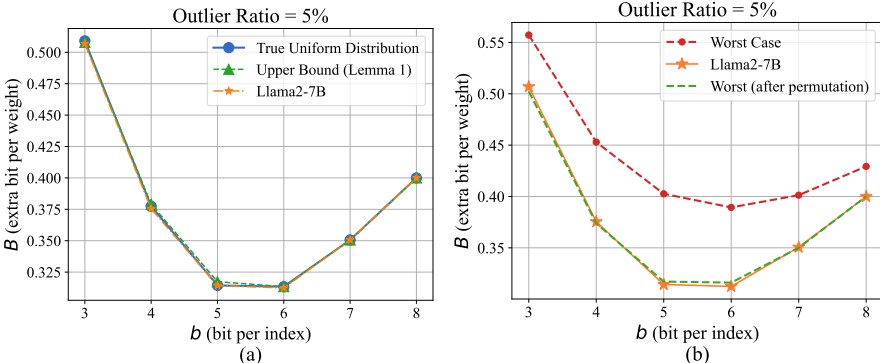

Figure 4: (a) ICQuant requires $B \approx 0.31$ bits/weight with $b = 6$ and $\gamma = 5\%$; (b) ICQuant adapts to any outlier distributions with graceful change in storage overhead.

Lemma 1 not only provides a theoretical guarantee on the efficiency of our index coding scheme, but also guides the optimal choice of $b$ with a given outlier ratio $\gamma$. In Figure 4 (a), we compare different choices of $b$ and the resulting storage overhead $B$ in three cases: (i) the upper bound calculated using Lemma 1, (ii) simulation using uniformly distributed outlier positions (synthetic) and (iii) empirical estimation using Llama2-7B. We notice that the three plots almost coincide, which confirms the uniform distribution observation and the tightness of our upper bound. Note that the convex behavior is due to the trade-off between two factors: (i) the base storage cost associated with each index; and (ii) the extra overhead caused by the index count accumulation of large gaps. Indeed, ICQuant supports a flexible range of outlier ratios that exhibit different trade-offs, with details in Appendix D.

In addition, although our index coding scheme is inspired by the uniform distribution property, it remains compatible with any outlier position distribution - the distribution only affects the storage efficiency. The worst case occurs, for example, when outliers cluster at the beginning and end of a weight channel, requiring additional flag values to encode the large gaps in between. Lemma 2 provides a universal upper bound on the total storage overhead to encode outlier positions, regardless of the underlying distribution. The proof, including details on the worst-case scenario, is provided in Appendix A.2.

**Lemma 2** *For a weight $\boldsymbol{w} \in \mathbb{R}^{d_{in}}$ with $\gamma d_{in}$ outliers, and for any distribution of outlier positions, if $b$ bits are used to encode each index gap, the total storage overhead B (defined as the number of bit per weight to store the outlier positions) satisfies*

$$B \leq \left( \frac{(1 - \gamma + \frac{1}{d_{in}})}{2^b - 1} + \gamma \right) b.$$

In Figure 4 (b), we compare the worst-case storage cost against the empirical estimation using Llama2-7B. Although the worst-case distribution incurs more storage overhead, the difference is less than 0.1 bit per weight. Furthermore, when outliers are not uniformly distributed, we can perform a one-time random permutation as described in Appendix C.2, noting that such a permutation does not incur additional computational overhead. As shown in Figure 4 (b), this random permutation nearly eliminates the gap.

## 4 Experiments

**Models and Datasets.**
We evaluate our method on the Llama2 and Llama3 family of models (Touvron et al., 2023; Grattafiori et al., 2024), across a wide range of scales from 1B to 70B parameters. Following recent PTQ work (Frantar et al., 2022; Chee et al., 2023; Tseng et al., 2024a; Malinovskii et al., 2024), we report perplexity on WikiText-2 (Merity et al., 2016) and C4 (Raffel et al., 2020) validation sets, as well as zero-shot accuracy on WinoGrande (Sakaguchi et al., 2021),

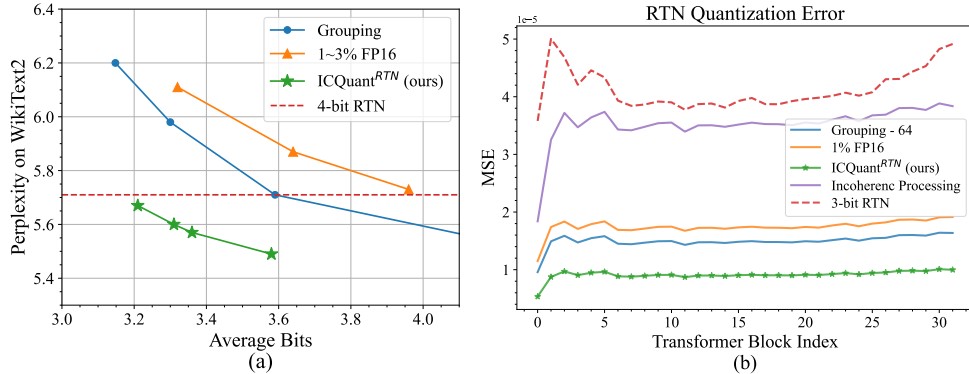

Figure 5: (a) Wikitext2 perplexity (↓) and (b) the quantization error (MSE) of 3-bit rounding-to-nearest (RTN) uniform quantization using different outlier suppression techniques, where the MSE is averaged across all linear layers in each transformer block.

PiQA (Bisk et al., 2020), ARC-easy, and ARC-challenge (Clark et al., 2018). The zero-shot accuracies are evaluated using LM Eval Harness v0.3.0 (Gao et al., 2024).

**Baseline Methods.**
We compare against the state-of-the-art weight-only algorithms, including *scalar quantization* algorithms, OmniQuant (Shao et al., 2023), SqueezeLLM (Kim et al., 2023) and QuIP (Chee et al., 2023), as well as *vector quantization* algorithms QuIP# (Tseng et al., 2024a), AQLM (Egiazarian et al., 2024), and QTIP (Tseng et al., 2024b). Another recent work, PV-Tuning (Malinovskii et al., 2024), builds on AQLM and focuses on optimizing the fine-tuning strategy. Since we do not perform any fine-tuning, we do not use it as a comparison in this paper. Note that fine-tuning can be done on top of our method and is left for future work.

**Choice of Quantizers.**
As mentioned before, our proposed quantization framework ICQuant can be applied with any quantization scheme. We here implemented two simple *scalar quantization* schemes that typically show significant performance degradation in extreme quantization regimes (2-3 bits per weight): (i) RTN: rounding-to-nearest uniform quantization and (ii) SK: sensitivity-aware K-means clustering (Kim et al., 2023). We refer to the overall quantization methods as ICQuant[RTN] and ICQuant[SK].

### 4.1 Compare Outlier Suppression Techniques.

To verify the effectiveness of ICQuant (described in Section 3) in outlier suppression, we compare it with the following techniques:

- **Grouping**: Divide the weights into small continuous blocks and quantize each group separately.
- **Mixed-Precision**: Keep outliers in full precision (FP16).
- **Incoherence Processing (Chee et al., 2023)**: Apply random orthogonal matrices to both sides of the weight matrices before quantization.

We first evaluate these outlier suppression techniques on 3-bit rounding-to-nearest (RTN) quantization. Figure 5 (a) shows the Wikitext2 perplexity as a function of the average number of bits per weight, where we adjust the hyperparameters (e.g. group size, outlier ratio) to achieve the different bitrates. Although grouping, mixed-precision, and ICQuant all improve quantization quality at the cost of extra storage, ICQuant[RTN] demonstrates the best trade-off between performance and storage efficiency. Notably, ICQuant[RTN] surpasses 4-bit RTN performance with less than 3.2 bits per weight. In addition, we found that incoherence processing on weights alone offers minimal benefits in extreme quantization scenarios, with the resulting perplexity falling outside the range reported in Figure 5 (a).

| Model | Method | bits | (ctx. 2048) Wiki2↓ | C4↓ | (ctx. 4096) Wiki2↓ | C4↓ |
|---|---|---|---|---|---|---|
| Llama2 - 7B | FP16 | 16 | 5.47 | 6.97 | 5.12 | 6.63 |
| | SqueezeLLM - 0.45% | 2.2 | 10.79 | - | - | - |
| | OmniQuant - g64 | 2.3 | 9.62 | 12.72 | - | - |
| | QuIP | 2 | - | - | - | - |
| | ICQuant$^{SK}$ - 5% | 2.3 | **7.21** | **8.97** | **6.75** | **8.84** |
| Llama2 - 13B | FP16 | 16 | 4.88 | 6.46 | 4.57 | 6.05 |
| | SqueezeLLM - 0.45% | 2.2 | 7.91 | - | - | - |
| | OmniQuant - g64 | 2.3 | 7.56 | 10.05 | - | - |
| | QuIP | 2 | - | - | 13.5 | 16.2 |
| | ICQuant$^{SK}$ - 5% | 2.3 | **6.10** | **7.80** | **5.74** | **7.57** |
| Llama2 - 70B | FP16 | 16 | 3.31 | 5.52 | 3.12 | 4.97 |
| | SqueezeLLM - 0.45% | 2.2 | 4.99 | - | - | - |
| | OmniQuant - g64 | 2.3 | 6.11 | 7.88 | - | - |
| | QuIP | 2 | - | - | 5.90 | 8.17 |
| | ICQuant$^{SK}$ - 5% | 2.3 | **4.26** | **6.22** | **4.01** | **5.78** |

Table 2: Wikitext2 and C4 perplexity (↓) of all Llama2 models quantized to 2-bit regime using *scalar quantization* algorithms. We use different context lengths (ctx.) to match the settings used in their original papers; blank entries indicate results that were not reported by the original authors.

In Figure 5 (b), we further compare the quantization error (i.e., $\|\mathcal{Q}(w) - w\|_2^2$) of $\approx 3.3$-bit RTN quantization using different outlier suppression techniques. We adjust the hyperparameters for all methods (apart from incoherence processing) to share similar extra storage overhead. The results show that ICQuant$^{RTN}$ leads to consistently lower error across the whole model. Specifically, ICQuant$^{RTN}$ reduces the quantization error to approximately 1/4 of the baseline, which verifies our intention of halving the quantization range by separating 5% outliers. Similarly, we found that incoherence processing yields little improvement. This is because when the weights already mostly follow a normal distribution, the random rotation does not effectively reduce the quantization range, as discussed in Appendix H.2.

Next, we compare (in Table 2) the performance of ICQuant$^{SK}$ to the state-of-the-art scalar quantization algorithms supported by the aforementioned outlier suppression techniques in the 2-bit regime. In particular, ICQuant$^{SK}$ and SqueezeLLM use the same quantizer (i.e., the sensitivity-aware K-means clustering), but SqueezeLLM (Kim et al., 2023) keeps outliers in FP16. OmniQuant (Shao et al., 2023) combines grouping with learnable clipping ranges. QuIP (Chee et al., 2023) applies incoherence processing and adaptive rounding. Table 2 shows that ICQuant$^{SK}$ significantly outperforms other algorithms in all models, with similar storage overhead.

## 4.2 Compare with SoTA weight-only PTQ

Finally, we conduct a comprehensive evaluation of ICQuant$^{SK}$ on different models and bitrates, comparing our results with the best-known baselines (AQLM, QuIP#, and QTIP) in weight-only PTQ. It is worth noting that all these baseline methods use complex vector quantization schemes and fine-tuning during/after quantization. Vector quantization typically involves significantly higher complexity in both quantization and inference, while fine-tuning requires substantial time and computational resources. In contrast, ICQuant$^{SK}$ aims to achieve their performance levels using a simple scalar quantizer with much less calibration data and no fine-tuning. The ablation study on the choice of outlier ratio and the improvements on the vanilla SK quantizer can be found in Appendix F. More implementation details are given in Appendix E.

Table 3 shows the perplexity and zero-shot accuracy of quantized Llama3-70B models (results of other Llama3 models are provided in Appendix G.). Since baseline models only report results with fine-tuning, we compare against their fine-tuned versions. Llama3-70B is known to be more challenging to quantize than Llama2 models (Huang et al., 2024), with state-of-the-art methods (QuIP# and QTIP) showing significant performance degradation

in downstream tasks, even after fine-tuning. Remarkably, ICQuant$^{SK}$ outperforms these methods by a large margin. For example, with 0.31 bits extra overhead, ICQuant$^{SK}$ improves QTIP's 2-bit accuracy on ARC-challenge from 28% to 50.7% and on ARC-easy from 35.3% to 81.7%, even surpassing QTIP's 3-bit performance. Interestingly, while QuIP# and QTIP achieve low perplexity, they perform poorly on zero-shot reasoning tasks, likely due to overfitting during fine-tuning. Moreover, ICQuant$^{SK}$ even achieves comparable zero-shot accuracies to PV-tuning, which fine-tunes vector quantization with an improved but more expensive fine-tuning strategy with complexity comparable to QAT.

Table 4 compares the perplexity of quantized Llama2 families. Overall, ICQuant$^{SK}$ achieves the best performance without fine-tuning. Compared to fine-tuned baselines, ICQuant$^{SK}$ consistently outperforms AQLM, reaching results comparable to QuIP# and QTIP. In addition, we observe that separating more outliers (e.g., 8.25%) improves the performance by further shrinking the range of inliers while quantizing more coarsely the outliers. This improvement occurs not only because of the larger proportion of inliers, but also because the inliers tend to be more important (see Appendix H.1). We report the evaluation on zero-shot reasoning tasks in Appendix G.

| Method | bits | Llama3 - 70B | | | | | |
| --- | --- | --- | --- | --- | --- | --- | --- |
| | | Wiki2↓ | C4↓ | ArcC↑ | ArcE↑ | PiQA↑ | Wino↑ |
| BP16 | 16 | 2.59 | 5.78 | 60.2 | 86.9 | 82.3 | 80.6 |
| QuIP# | 4.0 | [2.99] | [5.96] | [35.0] | [67.3] | [71.9] | [76.7] |
| QTIP | 4.0 | [2.75] | [5.83] | [56.1] | [83.9] | [81.3] | [80.6] |
| ICQuant$^{SK}$-5% | 4.3 | **2.72** | **5.83** | 59.6 | **87.2** | **82.6** | 80.4 |
| QuIP# | 3.0 | [3.59] | [6.18] | [31.1] | [36.6] | [58.8] | [76.4] |
| QTIP | 3.0 | [3.18] | [5.98] | [48.6] | [77.8] | [77.8] | [79.7] |
| ICQuant$^{SK}$-5% | 3.3 | 3.24 | 6.03 | 56.6 | **84.9** | **81.9** | **80.0** |
| QuIP# | 2.0 | [5.77] | [7.46] | [18.3] | [32.2] | [54.7] | [68.9] |
| QTIP | 2.0 | [4.97] | [6.80] | [28.0] | [35.2] | [57.1] | [72.6] |
| ICQuant$^{SK}$-8.25% | 2.4 | 5.83 | 8.22 | **52.0** | **82.3** | **79.7** | 74.0 |
| ICQuant$^{SK}$-5% | 2.3 | 5.65 | 7.64 | 50.7 | **81.7** | **79.4** | 75.5 |
| PV-tuning | 2.1 | [4.55] | [6.54] | [50.8] | [80.2] | [79.2] | [78.1] |

Table 3: Perplexity (↓) and zero-shot accuracy (↑) of Llama3-70B models (context length = 8192), quantized to 2-4 bit regime, using *vector quantization* algorithms and ICQuant$^{SK}$ (*scalar quantization*), where the values after fine-tuning are wrapped in [·]. We highlight the cases where ICQuant$^{SK}$ outperforms all fine-tuned baselines.

| Method | bits | Llama2 - 7B | | Llama2 - 13B | | Llama2 - 70B | |
| --- | --- | --- | --- | --- | --- | --- | --- |
| | | Wiki2↓ | C4↓ | Wiki2↓ | C4↓ | Wiki2↓ | C4↓ |
| FP16 | 16 | 5.12 | 6.63 | 4.57 | 6.05 | 3.12 | 4.97 |
| AQLM | 4.0 | - [5.21] | - [6.75] | - [4.65] | - [6.14] | - [3.19] | - [5.03] |
| QuIP# | 4.0 | 5.22 [5.19] | 6.79 [6.75] | 4.65 [4.63] | 6.15 [6.13] | 3.18 [3.18] | 5.02 [5.02] |
| QTIP | 4.0 | 5.17 [5.17] | 6.71 [6.69] | 4.62 [4.61] | 6.10 [6.09] | **3.16** [3.16] | **5.00** [5.00] |
| ICQuant$^{SK}$-5% | 4.3 | **5.17** | **6.70** | **4.61** | **6.09** | **3.16** | **5.00** |
| AQLM | 3.0 | - [5.46] | - [7.08] | - [4.82] | - [6.37] | - [3.36] | - [5.17] |
| QuIP# | 3.0 | 5.60 [5.41] | 7.34 [7.04] | 4.90 [4.78] | 6.50 [6.35] | 3.41 [3.35] | 5.20 [5.15] |
| QTIP | 3.0 | 5.38 [5.28] | 6.99 [6.87] | **4.74** [4.69] | 6.28 [6.22] | **3.27** [3.26] | **5.09** [5.08] |
| ICQuant$^{SK}$-5% | 3.3 | **5.35** | **6.95** | 4.75 | **6.26** | 3.28 | 5.10 |
| AQLM | 2.0 | - [6.59] | - [8.54] | - [5.60] | -[7.49] | - [3.94] | - [5.72] |
| QuIP# | 2.0 | 8.22 [6.19] | 11.0 [8.16] | 6.60 [5.35] | 8.07 [7.20] | 4.16 [3.91] | 6.01 [5.71] |
| QTIP | 2.0 | 6.82 [5.86] | 8.96 [7.73] | **5.52** [5.11] | 7.39 [6.85] | 3.87 [3.70] | 5.70 [5.48] |
| ICQuant$^{SK}$-8.25% | 2.4 | **6.35** | **8.25** | 5.54 | **7.25** | **3.86** | **5.61** |
| ICQuant$^{SK}$-5% | 2.3 | 6.75 | 8.84 | 5.74 | 7.57 | 4.01 | 5.78 |

Table 4: Wikitext2 and C4 perplexity (↓) of all Llama2 models (context length = 4096) quantized to 2-4 bit regime, using *vector quantization* algorithms and ICQuant$^{SK}$ (*scalar quantization*), where the perplexity values after fine-tuning are wrapped in [·]. We highlight in each case the best result without fine-tuning.

### 4.3 Inference Efficiency

Our outlier index decoding and weight dequantization procedure can be efficiently fused with matrix multiplication in a single kernel, thereby minimizing costly memory transactions. Alternatively, to fully accelerate the inference, outlier indices can be pre-decoded into a compact bitmask before the layer is executed. This offers flexibility depending on deployment constraints and latency requirements. To demonstrate the practical efficiency of ICQuant, we implemented an example 2-bit matrix-vector multiplication kernel, building on the open-sourced SqueezeLLM kernel. Our implementation supports on-the-fly decoding and de-quantization. In Table 5, we benchmark the matrix-vector multiplication subroutine on a Llama2-7B linear layer, using an Nvidia RTX 4090 GPU. The result shows that ICQuant can deliver faster inference than all baseline methods.

Although ICQuant requires additional decoding steps during inference, the overall latency remains lower than the baseline methods at comparable accuracy levels. Note that state-of-the-art baselines have non-trivial overhead: structured vector quantization schemes like AQLM require loading from a codebook that is too large to fit in cache, while QTIP and QuIP# require additional matrix multiplications and codebook generation. In addition, our example kernel serves as a proof-of-concept to demonstrate feasibility rather than peak performance. For instance, our current imple-

| Method | CUDA Latency |
|---|---|
| FP16 | 96.32 us |
| AQLM | 86.14 us |
| QuIP# | 31.28 us |
| QTIP | 32.92 us |
| ICQuant$^{SK}$-5% | 23.29 us |

Table 5: Latency of gate_proj layer in 2-bit Llama2-7B model.

mentation uses FP32 arithmetic, rather than FP16 (as used in the baselines), which does not fully leverage the tensor core acceleration available on modern GPUs.

## 5 Other Related Work

Model quantization has two main categories: post-training quantization (PTQ) and quantization-aware training (QAT) (Ma et al., 2024), depending on whether retraining is needed. Quantization can target weight-only, weight-activation quantization (Xiao et al., 2023; Liu et al., 2024a), and KV-cache (Liu et al., 2024b; Hooper et al., 2024). In this paper, we focus on weight-only PTQ. Initial research efforts focused on improving uniform quantization through innovative approaches. Notable contributions include error compensation introduced in GPTQ (Frantar et al., 2022) and equivalent scale transformations proposed in AWQ (Lin et al., 2024). Subsequently, Dettmers et al. (2023) and Kim et al. (2023) advanced the field using non-uniform quantization techniques, taking into account the weight sensitivity and statistical properties. Contemporary state-of-the-art methods Quip# (Tseng et al., 2024a), AQLM (Egiazarian et al., 2024), QTIP (Tseng et al., 2024b) have achieved remarkable results with 2-bit models using vector quantization and fine-tuning. Furthermore, PV-Tuning (Malinovskii et al., 2024) explored a new quantization-aware fine-tuning strategy for LLMs. Nevertheless, it is important to note that applying fine-tuning risks overfitting the calibration set and diminishes the efficiency advantage of PTQ over QAT.

## 6 Conclusions

In this paper, we present ICQuant, a novel method to decrease quantization range by separately quantizing outliers and efficiently encoding their indices — a method applicable to any quantization scheme. We demonstrate its effectiveness in improving weight quantization quality in extreme compression regimes ($\leq$ 4 bits), achieving state-of-the-art performance using a simple scalar quantizer and fixed outlier ratio. Looking ahead, promising directions include jointly optimizing the outlier ratio and storage overhead by leveraging layer-specific statistics, combining ICQuant with advanced quantization schemes and fine-tuning procedures, and extending the technique to quantize other components such as activations and KV-caches. Overall, ICQuant provides a flexible and highly effective mechanism for improving low-bit weight quantization, paving the way for more efficient deployment of LLMs under tight resource constraints.

## Acknowledgments

This work was supported in part by NSF grants #2221871, #2139304, #2146838, and the Army Research Laboratory grant under Cooperative Agreement W911NF-17-2-0196.

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

## Appendix

In this appendix, we provide further details as follows:

## A Proofs of Lemmas

### A.1 Proof of Lemma 1

**Lemma 1** *For a weight $w \in \mathbb{R}^{d_{in}}$ with $\gamma d_{in}$ outliers, if the weight values are randomly permuted (with uniform outlier positions) and $b$ bits are used to encode each index gap, the total storage overhead $B$ (defined as the number of bit per weight to store the outlier positions) satisfies*

$$\mathbb{E}_{uniform}(B) \leq \gamma b(1 + \frac{1}{e^{\gamma(2^b-1)} - 1}).$$

*Proof:* Recall that $\{i\}_{k=1}^{p}$ are the absolute indices of outliers, where $i_k \in [1, d_{in}]$ and $p = \lfloor \gamma d_{in} \rfloor$. We define the index gaps $x_0, x_1, \ldots, x_p$ as follows

$$x_0 = i_1, x_p = d_{in} - i_p + 1, \text{ and } x_k = i_{k+1} - i_k, \forall k = 1, \ldots, p-1.$$

Following the index coding scheme described in Section 3.2, we need to store $x_0, \ldots, x_{p-1}$ using $b$-bit representation and there are two possible cases:

(i) If $x_i \leq 2^b - 1$, we directly store $x_i$.

(ii) Otherwise, we store $\{2^b, 2^b, \ldots, (x_i - 1) \bmod (2^b - 1) + 1\}$, where $2^b$ is a flag value that means we accumulate $(2^b - 1)$ index count during decoding, and the number of such flags $2^b$ equals $\lfloor \frac{x_i-1}{2^b-1} \rfloor$.

Let $\mathcal{E}$ be the total number of empty intervals (i.e., the number of times the flag value $2^b$ is stored). Then we can write the expected total storage overhead as

$$\mathbb{E}(B) = \frac{bp}{d_{in}} + \frac{b}{d_{in}}\mathbb{E}(\mathcal{E}). \tag{1}$$

It remains to bound $\mathbb{E}(\mathcal{E})$. Note that $x_0, x_1, \ldots, x_p$ are positive and satisfy $\sum_{k=0}^{p} x_k = d_{in} + 1$. Moreover, by symmetry, any permutation of $\bar{x} = (x_0, x_1, \ldots, x_p)$ defines a valid distribution of the outlier positions with the same probability as $\bar{x}$. Hence, $x_i \sim x_j, \forall i \neq j$ (i.e., all $x_i$ share the same marginal distribution). Let $m = (2^b - 1)$. The expected number of empty intervals $\mathcal{E}$ can therefore be written as

$$\begin{aligned}
\mathbb{E}(\mathcal{E}) &= \sum_{k=0}^{p-1} \mathbb{E}(\lfloor \frac{x_k - 1}{m} \rfloor) \\
&\leq \sum_{k=0}^{p-1} \mathbb{E}(\lfloor \frac{x_k}{m} \rfloor) \\
&= p\mathbb{E}(\lfloor \frac{x_0}{m} \rfloor),
\end{aligned} \tag{2}$$

where $\mathbb{E}(\lfloor \frac{x_0}{m} \rfloor) = \sum_{j \geq 1} \mathbb{P}(\lfloor \frac{x_0}{m} \rfloor \geq j)$.

We observe that

$$\lfloor \frac{x_0}{m} \rfloor \geq j \iff x_0 \leq jm. \tag{3}$$

Therefore, we can write

$$
\begin{aligned}
\mathbb{E}(\lfloor \frac{x_0}{m} \rfloor) \quad &= \quad \sum_{j \geq 1} \mathbb{P}(x_0 \leq jm) \\
&= \quad \sum_{j \geq 1} \frac{\binom{d_{in}-jm}{p}}{\binom{d_{in}}{p}} \\
&\overset{\text{Claim 1}}{\leq} \quad \sum_{j \geq 1}(1 - \frac{jm}{d_{in}})^p \\
&\overset{(i)}{\leq} \quad \sum_{j \geq 1} e^{-\frac{jm}{d_{in}}p} \\
&\overset{(ii)}{\leq} \quad \sum_{j \geq 1} e^{-jm\gamma} \\
&= \quad \frac{1}{e^{m\gamma}-1} ,
\end{aligned}
\tag{4}
$$

where Claim 1 is proved after this lemma; (i) is due to the inequality that $1 - x \leq e^{-x}$ for any $x \in \mathbb{R}$; and (ii) follows $p = \lfloor \gamma d_{in} \rfloor$.

Finally, combining equation 1, equation 2 and equation 4, we have

$$\mathbb{E}(B) \leq \frac{bp}{d_{in}} + \frac{bp}{d_{in}} \cdot \frac{1}{e^{m\gamma}-1} \leq \gamma b + \gamma b \cdot \frac{1}{e^{2^b-1}-1}, \tag{5}$$

where the last inequality follows $p = \lfloor \gamma d_{in} \rfloor$ and $m = 2^b - 1$. $\qquad \square$

**Claim 1** *For $a, p, N \in \mathbb{Z}^+$, with $a < N$ and $p < N$, we have*

$$\frac{\binom{N-a}{p}}{\binom{N}{p}} \leq (1 - \frac{a}{N})^p. \tag{6}$$

*Proof:*

$$
\begin{aligned}
\frac{\binom{N-a}{p}}{\binom{N}{p}} &= \frac{(N-a)!(N-p)!}{(N-a-p)!N!} \\
&= \prod_{i=0}^{p-1} \frac{N-a-i}{N-i} \\
&= \prod_{i=0}^{p-1}(1 - \frac{a}{N-i}) \\
&\leq \prod_{i=0}^{p-1}(1 - \frac{a}{N}) \\
&= (1 - \frac{a}{N})^p
\end{aligned}
\tag{7}
$$

$\qquad \square$

## A.2 Proof of Lemma 2

**Lemma 2** *For a weight $w \in \mathbb{R}^{d_{in}}$ with $\gamma d_{in}$ outliers, and for any distribution of outlier positions, if $b$ bits are used to encode each index gap, the total storage overhead $B$ (defined as the number of bit per weight to store the outlier positions) satisfies*

$$B \leq \left( \frac{(1 - \gamma + \frac{1}{d_{in}})}{2^b - 1} + \gamma \right) b.$$

*Proof:* Let $\{i\}_{k=1}^{p}$ denote the absolute indices of the outliers, with $i_k \in [1, d_{in}]$ and $p = \lfloor \gamma d_{in} \rfloor$. We define $i_0 = 0$. When applying the index coding scheme proposed in Section 3.2, we store each index gap $i_k - i_{k-1}$ as $\mathcal{E}_k$ ($\mathcal{E}_k \geq 0$) number of flag values (i.e., $2^b$) and an overflow part $y_k$ ($1 \leq y_k \leq 2^b - 1$), such that $i_k - i_{k-1} = \mathcal{E}_k(2^b - 1) + y_k$. We need $b$ bits to store each $y_k$ or flag value, and the total storage overhead $B$ can be written as

$$B = \frac{b}{d_{in}} \left( \sum_{k=1}^{p} \mathcal{E}_k + p \right). \tag{8}$$

Now, it remains to upper bound the total number of flag values required $\sum_{k=1}^{p} \mathcal{E}_k$. Since $i_p \leq d_{in}$ and $i_0 = 0$, we have

$$\sum_{k=1}^{p} \left( \mathcal{E}_k(2^b - 1) + y_k \right) = i_p - i_0 \leq d_{in}, \tag{9}$$

which implies

$$\sum_{k=1}^{p} \mathcal{E}_k \leq \frac{d_{in} - \sum_{k=1}^{p} y_k}{2^b - 1} \leq \frac{d_{in} - p}{2^b - 1}, \tag{10}$$

where the second inequality is due to $y_k \geq 1$.

Combining (8) and (10), we can then upper bound the total storage overhead $B$ as follows:

$$\begin{aligned}
B &\leq \frac{b}{d_{in}} \left( \frac{d_{in} - p}{2^b - 1} + p \right) \\
&= \frac{b}{d_{in}} \left( \frac{d_{in} - \lfloor \gamma d_{in} \rfloor}{2^b - 1} + \lfloor \gamma d_{in} \rfloor \right) \\
&\leq \frac{b}{d_{in}} \left( \frac{d_{in} - \gamma d_{in} + 1}{2^b - 1} + \gamma d_{in} \right) \\
&\leq \left( \frac{(1 - \gamma + \frac{1}{d_{in}})}{2^b - 1} + \gamma \right) b.
\end{aligned} \tag{11}$$

$\square$

**Worst Case.** The upper bound in (10) is obtained when $i_p = d_{in}$ and $y_k = 1, \forall k = 1, \ldots, p$. One example of such worst cases is when outliers are concentrated at the end (and the beginning) of a weight channel (row). Specifically, for a weight vector $w$, the first $j$ ($0 < j < p$) elements and the last $p - j$ elements are outliers. In this case, the absolute outlier indices are

$$\{i\}_{k=1}^{p} = \{1, 2, \ldots, j, d_{in} - (p - j) + 1, \ldots, d_{in} - 1, d_{in}\},$$

with index gaps of 1 between consecutive indices, except for a single large gap in the middle.

# B  Range of Outliers

Similar to Figure 1 (a), we here provide plots of the normalized weight range taken by the outliers for other models in Llama2 and Llama3 Families in Figure 6.

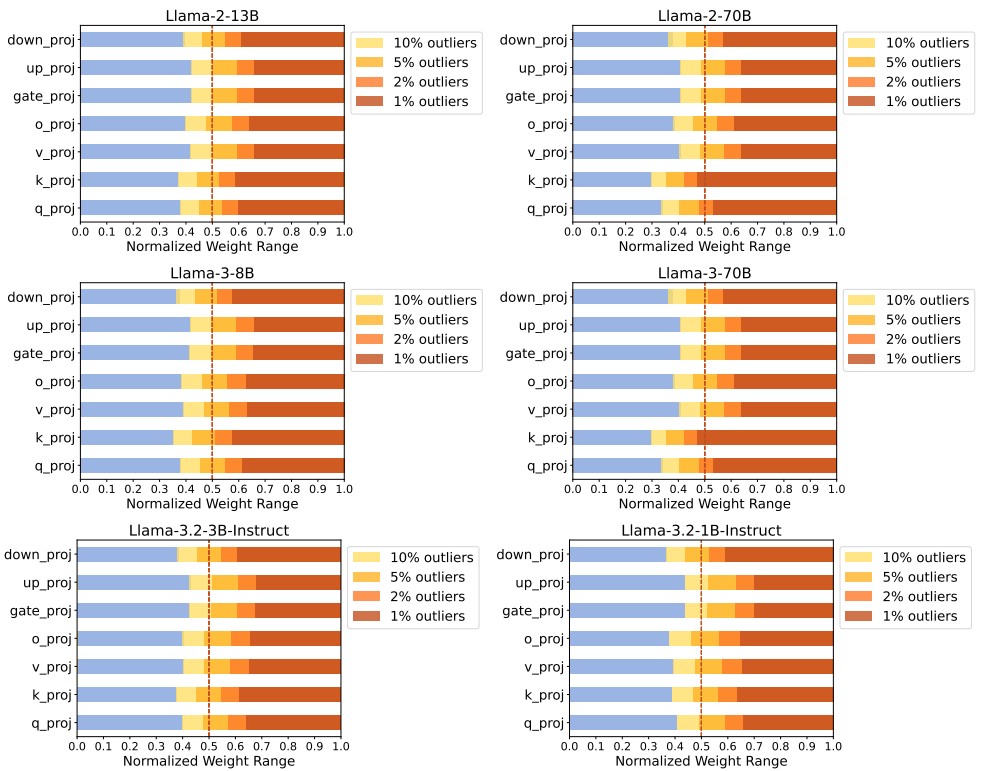

Figure 6: The normalized range taken by different amounts of outliers (starting from 1%), where the values of each type of layer are averaged over the whole model.

# C  Uniform Distribution of Outlier Positions

## C.1  Chi-square Test

We use a Chi-square goodness-of-fit test (Pearson, 1900) at a significance level of 0.05 to examine whether weight outliers are uniformly spread out across each output channel. Specifically, for each output channel (i.e., each row of the weight matrix), we divide the weight vector into non-overlapping groups of 256 elements. Assuming an outlier ratio of 6.25%, each group is expected to contain 16 outliers under the null hypothesis of uniform distribution. The Chi-square test is applied to each channel to test this hypothesis, and we report the rejection rate—the proportion of channels for which the null hypothesis is rejected—in Table 6. We observed consistent results across a wide range of popular LLMs, including Llama2 (Touvron et al., 2023), Llama3, Llama3.2-Instruct (Grattafiori et al., 2024), Llama4 Scout (Meta, 2025), and Qwen2.5-Instruct (Yang et al., 2024). This selection spans both base and instruction-tuned models, as well as dense and MoE architectures. Our results show that the positions of most weight outliers follow a uniform distribution, except for the output projection layers in self-attention blocks. In addition, we observe that within each model family, larger models show lower rejection rates.

| Model | | q_proj | k_proj | v_proj | o_proj | up_proj | gate_proj | down_proj |
|---|---|---|---|---|---|---|---|---|
| Llama 2 | 7B | 3.38% | 3.39% | 3.14% | 62.15% | 3.10% | 3.12% | 2.37% |
| | 13B | 2.95% | 2.98% | 2.91% | 59.37% | 2.96% | 2.96% | 2.16% |
| | 70B | 2.73% | 2.80% | 2.51% | 94.56% | 2.60% | 2.60% | 1.51% |
| Llama 3 | 8B | 3.01% | 3.13% | 2.95% | 95.25% | 3.10% | 3.08% | 2.11% |
| | 70B | 2.54% | 2.57% | 2.57% | 70.64% | 2.59% | 2.61% | 1.64% |
| Llama 3.2 | 1B | 3.49% | 3.94% | 3.66% | 82.33% | 3.45% | 3.45% | 2.65% |
| | 3B | 3.20% | 3.18% | 3.27% | 85.02% | 3.26% | 3.27% | 2.55% |
| Qwen 2.5 | 7B | 3.19% | 3.13% | 3.24% | 94.70% | 3.11% | 3.12% | 1.82% |
| | 32B | 2.77% | 2.73% | 2.98% | 90.37% | 2.78% | 2.80% | 1.56% |
| Llama 4 Scout | 17Bx16E | 3.09% | 3.06% | 3.13% | 97.37% | 2.99% | 2.99% | 2.60% |

Table 6: Chi-Square test rejection rates for different LLMs.

## C.2 Example of Random Permutation

As mentioned in Section 2, when outlier positions do not naturally follow a uniform distribution, we can enforce uniformity through a one-time random permutation before quantization. Given a weight matrix $W$, we aim to distribute outliers across each output channel (i.e., a row of the matrix). To achieve this, we randomly shuffle the columns in $W$ by right multiplication with a random permutation matrix $P$, producing $WP$. We then reorder the input (activation) $X$ by left multiplication with $P^\top$, yielding $P^\top X$. Since the permutation matrix $P$ is orthogonal, the linear transformation output remains unchanged: $WPP^\top X = WX$. Note that if $W$ is not the first linear transformation, the reorder of the input $X$ can be combined with the previous linear layer by permuting its output channels (rows of the weight matrix).

Figure 7 illustrates this multilayer permutation process using an MLP block from Llama's architecture, where $P_1$ and $P_2$ are random permutation matrices. Although the permutation for each layer can potentially be different, we use the same $P_1$ throughout to propagate the input order, accommodating the residual structure of the transformer. Note that these permutations preserve the model architecture because the permutation matrices can be absorbed into $W$ — we only need to store $P_1^\top W P_2$ (or $P_2^\top W P_1$). For the last linear layer in the model, we maintain the original order of its output channels, ensuring the model output remains unchanged.

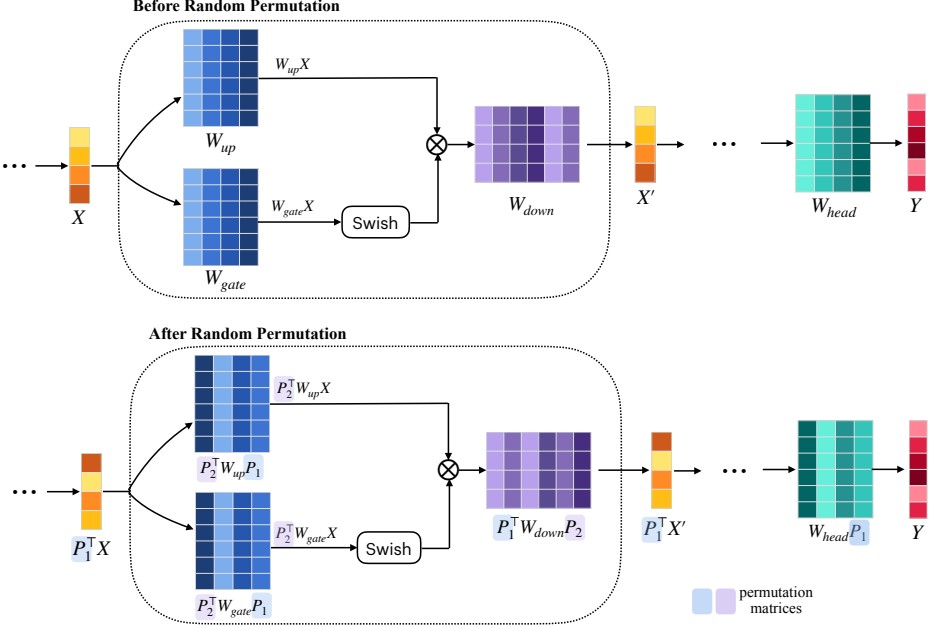

Figure 7: Example of random permutations for an MLP block in Llama-like transformers.

## D  Index Storage Cost Analysis

Figure 8 plots the storage overhead of ICQuant under different outlier ratios.

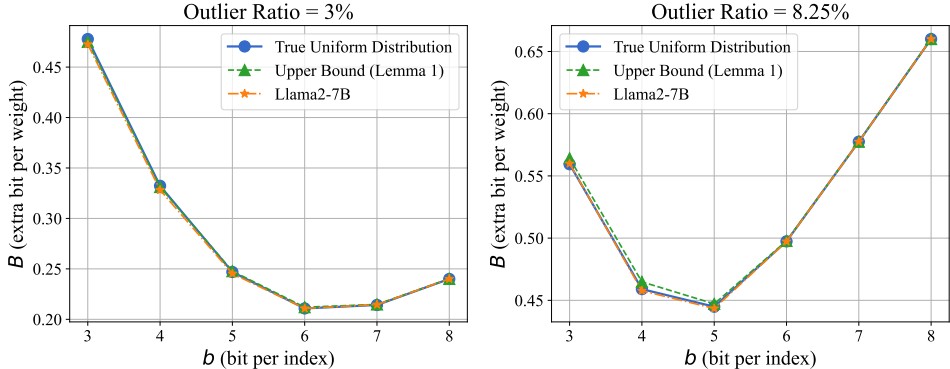

Figure 8: Index storage cost estimation of ICQuant.

## E  Implementation Details

### E.1  Quantizers

In the following we describe the implementation details of the two quantization schemes used in Section 4.

- ICQuant$^{\text{RTN}}$: We apply standard rounding-to-nearest (RTN) uniform quantization on the inlier weights that are centered around zero. Since the positive and negative outliers are separated on the two tails of the distribution, we use 1 bit to denote the sign and quantize them separately using $n - 1$ bits RTN.

- ICQuant$^{\text{SK}}$: Following SqueezeLLM (Kim et al., 2023), we apply weighted K-means clustering to minimize the proxy objective,

$$Q(\boldsymbol{w})^* = \arg\min_{Q}(W - W_Q)^\top \boldsymbol{H}(W - W_Q),$$

where the Hessian matrix $\boldsymbol{H}$ is approximated using the Fisher information matrix that is generated with 128 randomly selected sequences from the C4 dataset (Raffel et al., 2020). Each sequence has a length of 2048 for Llama2 and 8192 for Llama3. We apply such k-means clustering twice - once on the inlier weights and once on the outliers. For non-uniform quantization, we can quantize the negative and positive outliers together using $n$ bits.

### E.2  Evaluation

We follow the evaluation setup in recent PTQ literature (Frantar et al., 2022; Lin et al., 2024; Tseng et al., 2024b), and compare our results directly with those reported in the baseline methods' papers.

In particular, we use the same test split as GPTQ (Frantar et al., 2022) to calculate the perplexity on Wikitext2 and C4. For zero-shot evaluations, we use LM Eval v0.3.0 for all Llama2 models, matching the baseline methods. Note that the latest version of the library (v0.4.x) produces different results (typically $> 2\%$ difference), while even within the same version there is minor variance ($< 0.5\%$).

### E.3 Inference Kernel

At runtime, we first decode the outlier indices and dequantize them to full precision before multiplying them with the unquantized activation vectors. To minimize global memory access and kernel launch overhead, we implement decoding and dequantization on the fly and fuse them with matrix multiplication. However, since the decoding step is inherently sequential and difficult to parallelize on GPUs, we encode outlier gaps within smaller blocks (e.g., 256 columns). This approach enables parallel decoding of each block by a single thread. While the number of outliers per block may vary, we can store this information as a small auxiliary value. Due to the approximate uniformity of outlier positions, the variation of this number remains minimal, resulting in negligible additional storage overhead.

To isolate the impact of quantization algorithms from unrelated system-level factors, we benchmark GPU latency using a standard linear layer (mlp.gate_proj) from the LLaMA2-7B model. All latency measurements are conducted using the PyTorch CUDA profiler.

## F Ablation Study

### F.1 Outlier Ratio

In our experiments, we initiate with a 5% outlier ratio based on our empirical observation (Figures 1 and 6) that approximately 5% of outliers account for half of the total range. This separation effectively splits the weight range in half, ensuring consistent quantization resolution across all weights (both inliers and outliers).

To validate the impact of different outlier ratios, we evaluate ICQuant with 3-bit RTN uniform quantization on Llama2-7B, with outlier ratios ranging from 1% to 10%. Table 7 shows that the perplexity score decreases (indicating improved performance) as the outlier ratio increases, which aligns with our expectation since a smaller quantization range for inliers benefits the substantial majority (95%) of weights. However, we observed that the performance gain from separating additional outliers saturates beyond 5%. This is consistent with the Gaussian-like distribution of model weights, where range variations become negligible beyond this point. We therefore maintain a 5% outlier ratio throughout most experiments to balance between storage overhead and quantization quality.

| Outlier Ratio | 1% | 2% | 3% | 4% | 5% | 6% | 7% | 8% | 9% | 10% |
|---|---|---|---|---|---|---|---|---|---|---|
| Wiki2 Perplexity | 6.27 | 5.77 | 5.67 | 5.64 | 5.59 | 5.57 | 5.57 | 5.57 | 5.56 | 5.55 |

Table 7: Wikitext2 perplexity ($\downarrow$) of quantized Llama2-7B using ICQuant$^{\text{RTN}}$, with different outlier ratios.

### F.2 Compare with Vanilla SK Quantizer

In our main experiments, we apply ICQuant to the sensitivity-aware K-means (SK) quantizer proposed in the SqueezeLLM paper (Kim et al., 2023). By separating outliers and inliers during quantization, ICQuant effectively gains an additional bit of quantization capacity. Using only $\approx$2 bits, we expect ICQuant to achieve performance comparable to vanilla SK's 3-bit result, which outperforms all 2-bit state-of-the-art baselines. To demonstrate this, we present WikiText2 perplexity scores for Llama2-7B quantized at 2-3 bits in Table 8. ICQuant substantially improves upon 2-bit vanilla SK, approaching the 3-bit performance level. The remaining gap between 2-bit ICQuant and 3-bit vanilla SK exists because the non-uniform quantization biases toward the more concentrated inlier region. This finding also confirms that increasing the outlier proportion (e.g., to 8.25%) can further enhance model performance.

| Method | bits | Llama2 - 7B Wiki2↓ |
|---|---|---|
| FP16 | 16 | 5.12 |
| **Vanilla SK** | 3.0 | 5.71 |
| QuIP# | 2.0 | 8.22 |
| QTIP | 2.0 | 6.82 |
| ICQuant[SK]-8.25% | 2.4 | 6.35 |
| ICQuant[SK]-5% | 2.3 | 6.75 |
| **Vanilla SK** | 2.0 | 38.05 |

Table 8: Wikitext2 perplexity (↓) of quantized Llama2-7B using ICQuant[SK] and Vanilla SK.

# G   Other Experiment Results

Across a wide range of LLMs, from 1B to 70B, ICQuant[SK] (scalar quantization) matches and sometimes exceeds the state-of-the-art (*vector quantization*) performance.

| Method | bits | Llama2 - 7B | | | | Llama2 - 13B | | | | Llama2 - 70B | | | |
|---|---|---|---|---|---|---|---|---|---|---|---|---|---|
| | | ArcC↑ | ArcE↑ | PiQA↑ | Wino↑ | ArcC↑ | ArcE↑ | PiQA↑ | Wino↑ | ArcC↑ | ArcE↑ | PiQA↑ | Wino↑ |
| FP16 | 16 | 40.0 | 69.3 | 78.4 | 67.2 | 45.6 | 73.2 | 78.8 | 69.6 | 51.1 | 77.7 | 81.1 | 77.0 |
| AQLM | 4.0 | [41.0] | [70.2] | [78.2] | [67.3] | [44.8] | [73.3] | [78.4] | [69.9] | [50.7] | [77.3] | [81.5] | [76.5] |
| QuIP# | 4.0 | [40.4] | [68.6] | [78.5] | [67.4] | [43.6] | [71.3] | [78.7] | [69.6] | [50.5] | [77.7] | [81.4] | [77.3] |
| QTIP | 4.0 | [40.4] | [68.9] | [78.4] | [67.1] | [44.8] | [73.6] | [78.9] | [69.9] | [50.0] | [77.8] | [81.3] | [76.9] |
| ICQuant[SK]-5% | 4.3 | 40.5 | 69.0 | 78.1 | **67.4** | 45.6 | 72.5 | 78.8 | 69.1 | 50.6 | **77.9** | 81.1 | 76.6 |
| AQLM | 3.0 | [38.4] | [68.1] | [76.9] | [66.9] | [42.6] | [70.9] | [77.3] | [68.4] | [50.0] | [77.6] | [81.3] | [77.2] |
| QuIP# | 3.0 | [39.2] | [68.4] | [77.3] | [66.5] | [44.0] | [72.5] | [78.4] | [69.1] | [50.9] | [77.6] | [81.4] | [76.1] |
| QTIP | 3.0 | [38.9] | [68.1] | [78.1] | [66.9] | [44.0] | [72.8] | [78.0] | [69.5] | [50.0] | [78.2] | [80.6] | [77.0] |
| ICQuant[SK]-5% | 3.3 | 39.0 | 66.6 | 77.7 | **66.9** | 44.5 | 70.9 | **78.7** | **69.8** | 49.7 | 77.9 | 81.1 | 76.0 |
| AQLM | 2.0 | [32.8] | [63.7] | [74.8] | [65.7] | [38.8] | [69.3] | [75.9] | [68.8] | [47.9] | [77.7] | [80.4] | [75.9] |
| QuIP# | 2.0 | [35.2] | [65.3] | [75.4] | [64.9] | [39.6] | [69.0] | [77.3] | [67.4] | [47.6] | [77.1] | [79.5] | [74.6] |
| QTIP | 2.0 | [35.7] | [65.6] | [75.9] | [64.7] | [41.4] | [70.8] | [77.3] | [67.6] | [48.0] | [76.3] | [80.2] | [75.1] |
| ICQuant[SK]-8.25% | 2.4 | **35.9** | 63.6 | **76.1** | 65.6 | 39.8 | 68.4 | 77.0 | 64.2 | **48.6** | 75.2 | **81.3** | 74.7 |
| ICQuant[SK]-5% | 2.3 | 34.1 | 62.0 | 75.0 | 63.6 | 39.4 | 69.1 | 76.9 | 64.7 | 46.2 | 74.0 | 80.2 | 74.8 |

Table 9: Zero-shot accuracy (↑) of all Llama2 models using *vector quantization* algorithms and ICQuant[SK] (*scalar quantization*), where the accuracy values after fine-tuning are wrapped in [·]. We highlight the cases where ICQuant[SK] outperforms all fine-tuned baselines.

| Method | bits | Llama3 - 8B | | | | | | |
|---|---|---|---|---|---|---|---|
| | | Wiki2↓ | C4↓ | ArcC↑ | ArcE↑ | PiQA↑ | Wino↑ |
| BP16 | 16 | 5.54 | 7.10 | 50.4 | 80.1 | 79.7 | 72.5 |
| QuIP# | 4.0 | [5.81] | [7.32] | [50.2] | [79.7] | [79.7] | [73.1] |
| QTIP | 4.0 | [5.67] | [7.20] | [50.2] | [79.6] | [79.4] | [73.4] |
| ICQuant[SK]-5% | 4.3 | 5.69 | 7.23 | 49.7 | **79.8** | **79.7** | 73.0 |
| QuIP# | 3.0 | [6.27] | [7.71] | [46.4] | [77.4] | [77.9] | [72.9] |
| QTIP | 3.0 | [6.01] | [7.48] | [49.2] | [79.3] | [79.2] | [74.5] |
| ICQuant[SK]-5% | 3.3 | 6.22 | 7.73 | 47.4 | 78.1 | 78.6 | 72.9 |
| QuIP# | 2.0 | [7.84] | [9.04] | [39.2] | [72.9] | [75.6] | [68.2] |
| QTIP | 2.0 | [7.33] | [8.62] | [44.2] | [75.2] | [77.6] | [70.7] |
| ICQuant[SK]-8.25% | 2.4 | 9.67 | 10.97 | 38.0 | 71.6 | 76.6 | 67.9 |
| ICQuant[SK]-5% | 2.3 | 11.53 | 12.78 | 35.8 | 70.7 | 74.9 | 66.6 |

Table 10: Perplexity (↓) and zero-shot accuracy (↑) of Llama3-8B models (context length = 8192), quantized to 2-4 bit regime, using *vector quantization* algorithms and ICQuant[SK] (*scalar quantization*), where the values after fine-tuning are wrapped in [·]. We highlight the cases where ICQuant[SK] outperforms all fine-tuned baselines.

| Model | Method | bits | Wiki2↓ | ArcC↑ | ArcE↑ | HellaSwag↑ | PiQA↑ | WinoGrande↑ |
|---|---|---|---|---|---|---|---|---|
| | BP16 | 16 | 11.57 | 35.9 | 68.5 | 45.2 | 74.3 | 59.7 |
| Llama3.2 - 1B | QTIP | 4.0 | [11.93] | [34.8] | [68.4] | [44.5] | [73.3] | - |
| | ICQuant$^{SK}$-5% | 4.3 | 11.96 | **35.7** | 68.1 | **44.8** | **73.9** | 59.7 |
| | BP16 | 16 | 9.58 | 43.3 | 74.3 | 52.2 | 75.7 | 67.3 |
| Llama3.2 - 3B | QTIP | 4.0 | [9.77] | [43.5] | [74.3] | [51.9] | [75.1] | - |
| | ICQuant$^{SK}$-5% | 4.3 | **9.73** | 42.1 | 74.1 | **52.0** | **75.1** | 66.8 |

Table 11: Perplexity (↓) and zero-shot accuracy (↑) of Llama3.2 instruction-tuned models (context length = 8192), quantized to 4-bit regime, using QTIP (*vector quantization*) and ICQuant$^{SK}$ (*scalar quantization*), where the values after fine-tuning are wrapped in [·]. For these smaller models, 4-bit ICQuant still achieves performance nearly identical to FP16, comparable to the state-of-the-art.

## H  Other Observations

### H.1  Outliers are less important

Although large-magnitude outliers in activations (i.e., input features) are consistently sensitive to quantization (Dettmers et al., 2022; Lin et al., 2024; Sun et al., 2024), weight outliers appear to behave differently. Interestingly, previous research has shown improvements through both clipping weight outliers (Shao et al., 2023) and by preserving them in full precision (Kim et al., 2023). To examine the importance of weight outliers, we measure their sensitivity scores using Fisher information as proposed in Kim et al. (2023). Figure 9 plots the sensitivity scores against the weight values of two representative weight channels in Llama2-7B. The result shows that weights in the distribution's tail regions have significantly lower sensitivity than those in the central region. This observation highlights the inefficiency of mix-precision approaches that preserve outliers in full precision. More importantly, it explains our experimental observations that separating more outliers - which enables more precise quantization of the critical inlier values - improves the quantization performance.

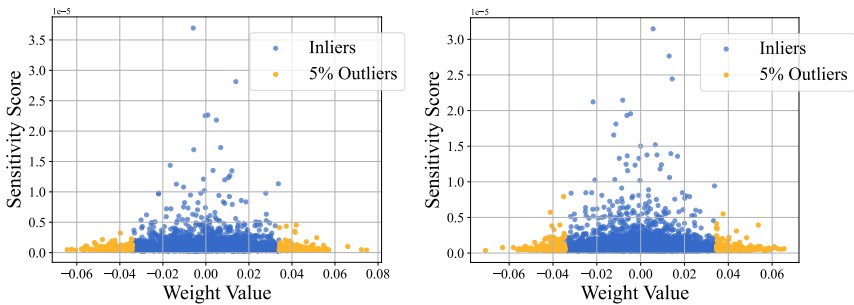

Figure 9: [Llama2-7B] Examples of weight values v.s. sensitivity scores.

### H.2  Examples of Incoherence Processing

To investigate the effect of incoherence processing (Chee et al., 2023) on reducing the quantization range, we visualized the weight distributions before and after applying random rotation, with examples provided in Figure 10 and Figure 11. We observed that incoherence processing can significantly reduce the weight range when there exist extremely large outliers, resulting in Gaussian-like distributions. However, the benefit of such rotation becomes negligible when the weight distribution already exhibits Gaussian-like behavior. The first case appears mostly in the initial layers of the model, which explains the results in Figure 5 (b) that the incoherence processing yields a substantial reduction in quantization MSE of the first transformer block, but shows small returns in subsequent blocks.

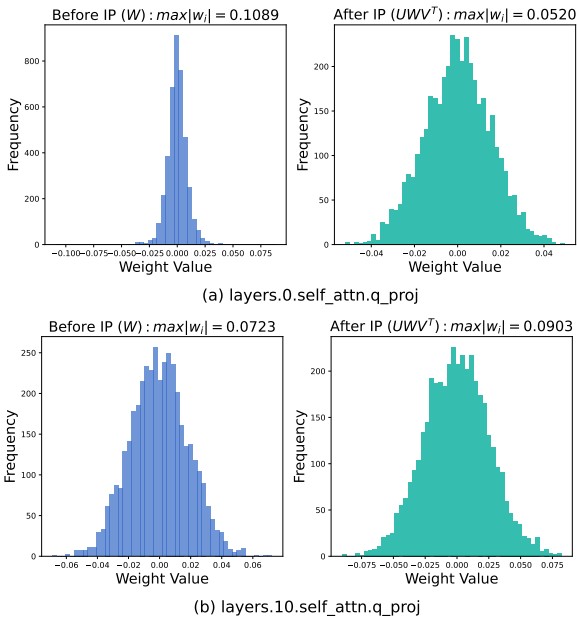

Figure 10: Examples of weight distribution in query_projection layers before/after incoherence processing.

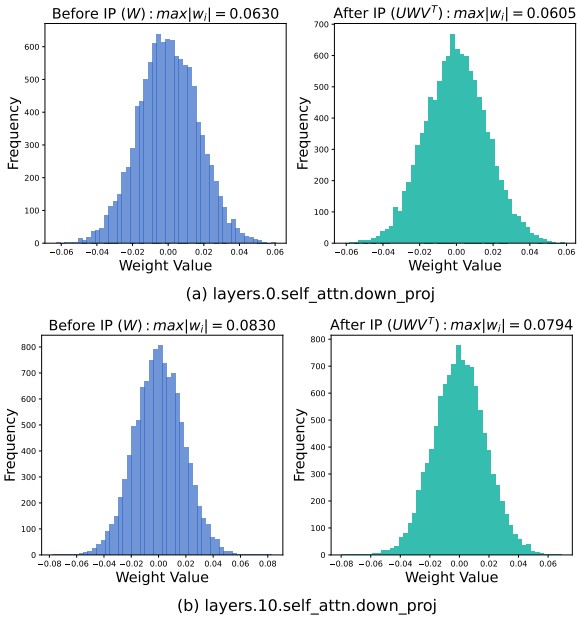

Figure 11: Examples of weight distribution in down_projection layers before/after incoherence processing.

