# OpenReview forum: "ICQuant: Index Coding enables Low-bit LLM Quantization"
_colmweb.org/COLM/2025/Conference — COLM 2025_

### Official Review · Reviewer_PnYh · 2025-05-05

**Rating:** 5
**Confidence:** 5
**Ethics Flag:** 1

**Summary:**

This paper introduces ICQuant, a quantization framework aimed at low-bit compression of LLMs. ICQuant specifically addresses the challenge of weight outliers with large magnitudes that inflate quantization ranges. Leveraging the observation that outliers distribute uniformly across weight matrices, ICQuant reduces the index storage cost to approximately 0.3 bits per weight. Experimental evaluations demonstrate that ICQuant improves quantization performance at extremely low bits, achieving SOTA performance without fine-tuning. This makes ICQuant particularly promising for deploying resource-constrained devices.

**Questions To Authors:**

1. Weight clipping has been shown to effectively manage weight outliers, as demonstrated in Sec. 3.3.2 of [LLMC](https://aclanthology.org/2024.emnlp-industry.12/). Could you provide a detailed comparison between your proposed method and this clipping approach?

2. The proposed method is promising, but practical implementation details related to hardware deployment are missing. Could you provide an example of an ICQuant GEMM kernel or relevant benchmarks to illustrate the practical efficiency of your method?

3. Could you report and discuss the performance of your quantized models on more challenging benchmarks, such as MMLU?

**Reasons To Accept:**

1. The paper is well written, well-structured, and easy to follow.
2. The paper introduces a novel approach to addressing the challenge posed by weight outliers, utilizing separate quantization ranges combined with an optimized indexing scheme.
3. The authors provide rigorous statistical analyses and theoretical bounds on their indexing scheme, ensuring methodological soundness and offering insights that may benefit future quantization techniques.
4. Experimental evaluations show that ICQuant substantially improves quantization performance and zero-shot accuracy over SOTA methods.

**Reasons To Reject:**

1. Limited generalizability of assumptions: The efficiency of ICQuant strongly depends on the assumption of a uniform distribution of outliers. While empirically validated in popular models, it remains unclear if this assumption is universally held, such as whether it is still suitable for the MoE structure models.
2. Complexity of practical implementation: Although the indexing scheme significantly reduces overhead theoretically, practical implementation complexities, such as hardware support and runtime performance implications, were not thoroughly analyzed. It is unclear if the theoretical advantages could translate into a practical, deployable kernel.
3. Insufficient analysis of computational cost: The proposed indexing scheme, while storage-efficient, introduces additional decoding complexity. The paper does not comprehensively evaluate the computational cost during inference.
4. Lack of ablation on hyperparameters: The paper chooses a fixed outlier ratio (e.g., 5%) for many experiments without extensive exploration or justification. More rigorous analysis or ablation studies on the sensitivity to hyperparameters would enhance confidence in their choice.

---

> ### Author Response · Authors · 2025-06-03
> **Responses to Reviewer PnYh**
>
> ### **Distribution Assumption**
> - To verify the uniform distribution property in MoE models, we performed the same Chi-square test on the Llama 4 Scout model (17Bx16E). The results align with other models.
>
> ||q|k|v|o|up|gate|down|
> |-|-|-|-|-|-|-|-|
> |rejection ratio (%)|3.09|3.06|3.13|97.37|2.99|2.99|2.60|
>
> - ICQuant could also apply even if this uniform distribution property does not hold. Please refer to our common response in the beginning for a detailed analysis.
> ### **Complexity & Computational Cost**
> - Our indices decoding can be efficiently fused with matrix multiplication in a single kernel, thereby minimizing overhead and avoiding costly memory transactions. Alternatively, to fully accelerate the inference, outlier indices can be pre-decoded into a compact bitmask before the layer is executed.
> - To demonstrate the practical efficiency of ICQuant, we implemented an example 2-bit matrix-vector multiplication kernel. Our implementation supports on-the-fly decoding and dequantization. We benchmark the matrix-vector multiplication subroutine on a standard linear layer, using an Nvidia RTX 4090 GPU. ICQuant can indeed deliver faster inference than all baseline methods.
>
> ||CUDA Latency (us)|
> |-|-|
> |FP16|96.3|
> |AQLM|86.1|
> |QuIP#|31.3|
> |QTIP|32.9|
> |ICQuant|23.3|
> - Although ICQuant requires additional decoding steps, the inference latency remains lower than baseline methods. Note that SOTA baselines have non-trivial overhead—structured vector quantization schemes like AQLM require loading from a codebook larger than the cache size, while QTIP and QuIP# require additional matrix multiplications and codebook generation.
> - Our kernel serves as a proof-of-concept to show feasibility rather than peak performance. For instance, unlike all baselines' implementations, our current kernel uses full-precision (FP32) arithmetic instead of FP16, which does not leverage tensor core acceleration available on modern GPUs. We believe there is significant room for further optimization, yet already the current results are very promising.
> ### **Hyperparameters**
> - We choose 5% outlier ratio for the following reasons:
>     - We split the weight range in half to ensure consistent quantization resolution for inliers and outliers.
>     - Our empirical analysis (Figure 1, 6) shows that approximately 5% of outliers account for half of the total range.
> - To verify the impact of outlier ratios, we evaluate ICQuant on 3-bit uniform quantization of Llama2-7B below, with varying outlier ratios.
>     - The perplexity decreases as the outlier ratio increases. This is expected, since a smaller quantization range for inliers benefits the majority of weights.
>     - However, the performance gain from separating more outliers saturates after 5%. This aligns with the behavior of Gaussian-like distributions, where the change in range becomes minimal beyond 5%. To balance the trade-off between storage overhead and quantization quality, we stay with 5% for most experiments.
>
> |outlier_ratio (%)|2|3|4|5|6|8|10|
> |-|-|-|-|-|-|-|-|
> |wiki2 PPL|5.77|5.67|5.64|5.59|5.57|5.57|5.55|
> ### **Q1**
> - Weight clipping cuts all outliers to two predetermined thresholds at both ends, which improves quantization quality by reducing the range of inliers (the majority of weights). However, the effectiveness of clipping is limited due to the large quantization error for outliers. In contrast, ICQuant quantizes the outlier area using the same number of bits, enabling more aggressive shrinkage of the inlier range. When targeting the same inlier range for $b$-bit quantization, weight clipping produces $2^b$ times larger quantization error for outliers compared to ICQuant.
> - We compare ICQuant's performance against the results reported in the LLMC paper, using the same benchmarks. Under the same storage cost, ICQuant significantly outperforms weight clipping.
>
> ||| Llama2-7B||
> |-|-|-|-|
> ||bits|Avg. PPL|Avg. Acc.|
> |AWQ w/ asym. clip|2.3|13.26|48.8|
> |ICQuant|2.3|8.09|59.4|
> ### **Q2**
> Please refer to the complexity response above, with results for an example kernel.
> ### **Q3**
> We evaluate ICQuant on the 5-shot MMLU dataset across the Llama2 family. ICQuant achieves near-lossless performance at ~3 and ~4-bit levels. While 2-bit quantization remains robust on the 70B model, smaller models show larger degradation. We also compared against 2-bit AQLM and QuIP#—the only baseline results reported on MMLU. For fair comparison, we applied the same fine-tuning strategy as used in these baseline results. (Values in brackets are post-fine-tuning.) We found that fine-tuning substantially improved MMLU performance, allowing ICQuant to outperform SOTA baselines.
>
> ||bit|7B|13B|70B|
> |-|-|-|-|-|
> |FP16|16|45.9|55.2|68.8|
> |ICQuant-5%|4.3|44.8|55.4|68.5|
> |ICQuant-5%|3.3|44.0|53.6|68.1|
> |ICQuant-8.25%|2.4|34.8 [40.7]|47.2 [50.5]|64.1|
> |ICQuant-5%|2.3|30.3 [39.0]|43.0 [49.7]|63.3|
> |AQLM|2.0|[38.5]|[48.8]|[65.3]|
> |QuIP#|2.0|[36.8]|[50.0]|[65.3]|

---

> > ### Comment · Reviewer_PnYh · 2025-06-09
> >
> > Thank you for the detailed response and for sharing the additional experimental results. Most of my concerns have been resolved, especially in computational cost. Considering that different LLMs show different weight distributions, it would be beneficial to present more LLMs in the revision. Taking these into account, I will maintain the initial score.

---

> > ### Author Response · Authors · 2025-06-09
> >
> > We are glad that our response has resolved your concerns and questions. We would appreciate it if you could consider increasing your score in light of this.
> >
> > Regarding your comment on weight distributions across different LLMs:
> >
> > Weights in trained LLMs typically follow Gaussian-like distributions, as supported by prior work (e.g., [QLoRA](https://arxiv.org/abs/2305.14314)), with outliers significantly impacting quantization quality. Our analytical results (Lemma 1 in the paper, Lemma 2 in our common response) demonstrate that ICQuant remains efficient regardless of the specific outlier distribution. Furthermore, our empirical evaluations confirm the uniform distribution property of outlier indices across 10 widely used LLMs, including Llama 4 Scout. This ensures ICQuant's robustness and practical effectiveness across diverse models.
> >
> > If there is a particular model that you would like us to look at the weight distribution, we are happy to do so.

---

### Official Review · Reviewer_ykoc · 2025-05-05

**Rating:** 5
**Confidence:** 3
**Ethics Flag:** 1

**Summary:**

This paper presents ICQuant, a framework that leverages outlier statistics to implement an index coding scheme for weight-only quantization. Compared to some existing methods, ICQuant reduces the overhead to about 0.3 bits, which may offer advantages in low-bit (2–3 bits) compression scenarios. It is designed to be compatible with existing quantizers to potentially enhance quantization quality. Experimental results show that ICQuant improves the zero-shot accuracy of a 2-bit Llama3-70B model and achieves performance comparable to fine-tuned quantizers, despite not requiring fine-tuning.

**Questions To Authors:**

See above

**Reasons To Accept:**

1. The authors provide an empirical observation of a uniform distribution of outliers and propose methods that require fewer bits of overhead to handle long-tail distributions.
2. The included proofs enhance the transparency and interpretability of the proposed algorithms.
3. The quantized models achieve perplexity comparable to prior works on WikiText2 and other benchmarks.
4. Overall, the paper is well-written and easy to follow.

**Reasons To Reject:**

1. Why are methods like GPTQ and AWQ not included in the comparisons? These are popular baselines in LLM quantization and their absence weakens the evaluation.
2. A paper titled "Scaling Laws for Quantized LLMs" (https://arxiv.org/pdf/2411.17691) finds that low-bit quantization methods (e.g., GPTQ, AWQ) tend to fail on fully trained LLMs due to increasing quantization loss rather than decreasing it. Could you run similar experiments on the Pythia suite (across more than one model size) along their training trajectories to examine whether ICQuant exhibits the same weakness?
3. Table 2 lacks performance data for certain configurations, such as the entire row of QuIP results for Llama2-7B. This makes it difficult to assess how well ICQuant performs in comparison.
4. While Table 3 shows substantial improvements—up to ~50% on some LLM benchmarks using Llama3-70B. However, the same trend is not clearly reflected in Tables 6, 7, and 8 for smaller models (1B–13B). Additionally, the performance improvement on Llama2-70B (Table 6) appears significantly smaller than that on Llama3-70B. More experimental results are needed to determine whether ICQuant is agnostic to model architecture, size, and dataset. These factors may influence the statistical distribution of weights and thus affect quantization performance.

---

> ### Author Response · Authors · 2025-06-03
> **Responses to Reviewer ykoc**
>
> ### **GPTQ & AWQ**
> The state-of-the-art baselines we compared with (QTIP, QuIP#, AQLM, SqueezeLLM, and OmniQuant) consistently outperform GPTQ and AWQ. As an illustration, we provide the perplexity evaluation (context length = 2048) of the 2.3-bit Llama2 models below. Additional evaluation results can be found in the papers of our baseline methods.
>
> |  |  | **Llama2 - 7B** |  | **Llama2 - 13B** |  |
> | --- | --- | --- | --- | --- | --- |
> |  | **bits** | **Wiki2** | **C4** | **Wiki2** | **C4** |
> | GPTQ - g64 | 2.3 | 20.85 | 19.40 | 22.44 | 12.48 |
> | AWQ - g64 | 2.3 | 2.1e5 | 1.6e5 | 1.2e5 | 9.5e4 |
> | ICQuant - 5% | 2.3 | **7.21** | **8.97** | **6.10** | **7.80** |
>
> ### **Scaling Laws Paper**
> - Our paper focuses on post-training quantization, and all evaluations are conducted on fully trained LLMs. We believe that studying quantization across training trajectories is a valuable direction, but it is somewhat orthogonal to our current focus.
> - The state-of-the-art baselines (QTIP, QuIP#, AQLM), which are not evaluated in the “scaling laws” paper, do not fail in the low-bit regime. However, these methods all rely on complex vector quantization and expensive fine-tuning. Our experimental results demonstrate that ICQuant can significantly improve the quantization quality of simple scalar quantization schemes (similar to GPTQ and AWQ), achieving the state—of-the-art performance in low-bit quantization without any fine-tuning.
>
> ### **Missing Performance Entries**
> Prior works evaluate perplexity scores using two different context length configurations. In Table 2, we present ICQuant's performance under both configurations and compare directly with results from the original baseline papers. Some entries are missing because the original papers did not report certain results—for example, QuIP did not provide numbers for the Llama2-7B model.
>
> ### **Experiment Results**
> - Llama3-70B is known to be particularly challenging to quantize, with many existing solutions failing to achieve good performance even after fine-tuning. ICQuant, however, remains robust in this setting and significantly improves upon the state-of-the-art results.
> - For other models, the performance gaps between quantized baselines and non-quantized models (FP16/BP16 in tables) are already minimal (e.g., 0-3% for Llama2 family). Yet achieving such near-lossless performance typically requires complex vector quantization and expensive end-to-end fine-tuning. ICQuant, by contrast, enables simple scalar quantization to reach state-of-the-art performance without any fine-tuning.
> - We evaluate ICQuant on 6 different models from Llama2 and Llama3 families, with scales ranging from 1B to 70B parameters, covering all benchmarks used in the state-of-the-art baseline papers.
> - Weights in trained LLMs typically follow Gaussian-like distributions, as verified in prior works (e.g., [QLoRA](https://arxiv.org/abs/2305.14314)). While the outlier position distribution may slightly affect the storage efficiency of our index coding scheme, it does not impact ICQuant’s core advantage of halving the quantization range by storing a small amount of outliers. (Please refer to our common response in the beginning for a detailed analysis of the storage efficiency.)

---

> > ### Comment · Reviewer_ykoc · 2025-06-10
> >
> > Thank you for the response and I appreciate the effort to address the raised concerns. I'll maintain my score.

---

### Official Review · Reviewer_bufP · 2025-05-08

**Rating:** 5
**Confidence:** 4
**Ethics Flag:** 1

**Summary:**

This paper introduces ICQuant, a novel framework for efficient low-bit Large Language Model (LLM) quantization that leverages statistical properties of weight outliers. Major contributions include:
* Separating weights into outliers and inliers for separate quantization
* Designing an efficient index coding scheme based on the uniform distribution of outliers
* Providing theoretical upper bounds on the storage overhead required by their index coding approach

**Questions To Authors:**

1. **Adaptive outlier ratio selection**: Consider implementing mechanisms for varying outlier ratios by layer characteristics, fully leveraging weight distribution statistics.
2. **Integration with advanced quantization schemes**: Explore how ICQuant could complement more sophisticated quantization approaches and targeted fine-tuning methods to further enhance performance.
3. **Deeper theoretical analysis**: Strengthen the theoretical analysis of outlier distribution properties, particularly examining robustness when the uniform distribution assumption doesn't hold.

**Reasons To Accept:**

* The method is quantizer-agnostic and compatible with various existing quantization schemes
* Using only 2.3 bits per weight, ICQuant significantly improves quantization performance, substantially enhancing Llama model series accuracy compared to state-of-the-art methods like QTIP and QuIP#
* The index coding scheme is highly storage-efficient, requiring merely ~0.3 bits overhead per weight
* Comprehensive evaluation across multiple model families

**Reasons To Reject:**

1. **Limited novelty**: The approach of separating outliers and inliers for separate processing is not fundamentally novel, as similar concepts appear in existing works like SqueezeLLM. Dividing weights into two groups for separate quantization is essentially a specific case of existing group quantization methods.
2. **Distribution assumption issues**: The claim regarding uniform per-channel distribution of outliers needs more validation. Table 1 shows rejection rates as high as 95.25% for certain layers (e.g., o_proj), significantly contradicting the uniform distribution hypothesis. Authors only briefly mention these anomalies have "minimal impact on the overhead," but provide insufficient evidence.
3. **Limited innovation in coding scheme**: The proposed index coding scheme is essentially a basic data compression technique, limiting its innovativeness.
4. **Insufficient analysis of permutation impact**: Authors suggest using random permutations to enforce uniformity in non-uniform cases but fail to analyze potential impacts on model performance, particularly on forward propagation speed. Figure 7 indicates that activations require an additional matrix multiplication, potentially increasing inference latency.
5. **Lack of theoretical justification for parameter selection**: The assertion that 5% of outliers occupy 50% of the value range appears somewhat arbitrary, lacking theoretical support or systematic investigation of different proportions.
6. **Puzzling experimental results**: The demonstration that results are better than quantization methods using fine-tuning is difficult to comprehend. Additional analysis should be provided, and code should be open-sourced promptly.

---

> ### Author Response · Authors · 2025-06-03
> **Responses to Reviewer bufP**
>
> ### **Novelty**
> - Outliers pose a major challenge for low-bit weight quantization, leading to many proposed outlier suppression solutions. ICQuant is the first practical framework that can efficiently encode the outlier indices, allowing to quantize outliers and inliers using separate codebooks. It offers distinct advantages:
>     - Existing approaches like SqueezeLLM store outlier values and their indices in full precision. Due to the high storage cost per outlier, they can only handle up to 1% outliers. In contrast, ICQuant can process 5x more outliers with the same storage cost. This reduces the 2-bit quantization-induced degradation by 56% on average, for the same quantizer (measured using Table 2).
>     - Traditional group quantization simply groups adjacent weight entries without considering their statistics. Irregular grouping is challenging since group labels need to be stored for decoding. ICQuant innovates by fully leveraging weight statistics for both grouping and efficient storage.
> - As shown in Section 4.1, ICQuant substantially outperforms existing outlier suppression techniques (including full-precision outlier storage and grouping) under the same storage cost.
> ### **Distribution**
> - The impact of o_proj layers is minimal for two reasons:
>     - While their outlier indices deviate from a perfectly uniform distribution, the storage cost of ICQuant's coding scheme remains modest. The table below shows empirical layer-wise costs.
>     - O_proj layers make up just 8% of all quantized weights in Llama3-8B.
>
> ||q|k|v|o|up|gate|down|
> |-|-|-|-|-|-|-|-|
> |Llama3-8|0.31|0.31|0.31|**0.32**|0.31|0.31|0.31|
> - We emphasize that ICQuant can be applied to any outlier distribution. We provide a detailed analysis in our common response.
> ### **Coding Scheme**
> Our primary innovation is in leveraging the statistical property of outliers to design a lightweight yet robust index coding scheme. This approach yields significant storage savings and offers a universal benefit for LLM quantization. The simplicity of our method is an advantage, not a limitation—it enables efficient deployment across LLMs at scale.
> ### **Permutation**
> Our random permutation strategy does not add storage or computation overhead. In Figure 7, we use $P_1^\top X$ to denote the reordered activation rather than a matrix multiplication. $P_1^\top$ combines with the previous $W$ and is stored as $P_1^\top W$.
> ### **Parameters**
> The rationale for our parameter choices is as follows:
> - We split the range at 50% to ensure consistent quantization resolution of all weights (i.e., inliers and outliers).
> - Our empirical analysis (Figure 1, 6) shows that approximately 5% of outliers account for half of the total range.
> ### **Experiments**
> - By separating outliers and inliers, ICQuant effectively gains an extra bit of quantization capacity. In our paper, we apply ICQuant to the non-uniform quantizer from SqueezeLLM paper. Using only ~2 bits, we expect ICQuant to approach SqueezeLLM's 3-bit result, which outperforms all 2-bit SOTA baselines. As shown in the Llama2-7B quantization results below, ICQuant substantially improves upon 2-bit SqueezeLLM, approaching 3-bit performance. The small gap between ~2-bit ICQuant and 3-bit SqueezeLLM exists because the non-uniform quantization biases toward the more concentrated inlier region. This also confirms that increasing the outlier ratio (e.g., to 8.25%) can further enhance performance.
>
> ||bit|Wiki PPL|
> |-|-|-|
> |**SqueezeLLM**|**3.0**|**5.72**|
> |QuIP#|2.0| 8.22|
> |QTIP|2.0| 6.82|
> ICQuant-8.25%|2.4|6.35|
> |ICQuant-5%|2.3|6.75|
> |**SqueezeLLM**|2.0|35.49|
> - We plan to release the code upon paper acceptance. If the program chairs can provide a mechanism to do this anonymously, we are happy to share the code if the reviewer deems it necessary.
> ### **Q1**
> Adaptive outlier separation is a promising future direction. In this work, we use a fixed outlier ratio for two reasons:
> - Our empirical observations (Table 1, 6) show that layer-wise distribution statistics remain similar across the models we evaluated.
> - The choice of outlier ratio reflects a tradeoff between storage cost and model accuracy, and using a global value enables easier control of this balance during evaluation and deployment.
> ### **Q2**
> - ICQuant's core strength—halving the quantization range—is orthogonal to quantization schemes. Moreover, it offers the flexibility to design different schemes for outliers and inliers based on their distinct statistics, which we leave for future work.
> - We conducted preliminary experiments using the same fine-tuning procedure as QuIP#/QTIP for 2-bit quantization of Llama2-7B. Fine-tuning further enhances ICQuant's performance beyond the SOTA.
>
> ||bit|ArcC|ArcE|PiQA|Wino|
> |-|-|-|-|-|-|
> |FP16|16|40.0|69.3|78.4|67.2|
> |QuIP#|2.0| 35.5|65.3|75.4|64.9|
> |QTIP|2.0|35.7|65.6|75.9|64.7|
> |ICQuant-5%|2.3|36.3|66.7|76.6|64.8|
> ### **Q3**
> Please refer to our common response in the beginning for details.

---

> > ### Comment · Reviewer_bufP · 2025-06-10
> >
> > Thank you for the detailed rebuttal. Most of my concerns are addressed. I will keep my original scores.

---

### Official Review · Reviewer_BVJU · 2025-05-12

**Rating:** 5
**Confidence:** 2
**Ethics Flag:** 1

**Summary:**

The paper identifies that only 5% of outliers account for 50% of the magnitude and proposes a more efficient parameter storage scheme, achieving 1-bit information storage at a cost of only 0.3 bits.

**Questions To Authors:**

Refer to the above

**Reasons To Accept:**

1.  The paper presents an effective method to reduce the cost of model storage and loading, and it can be orthogonally applied to various other quantization techniques.

2.  Experiments demonstrate the effectiveness of the proposed method compared to approaches with similar cost.

**Reasons To Reject:**

1.  The paper suffers from poor writing, characterized by undefined symbols and unclear expressions. For instance, line 94 mentions "weights with high magnitude (i.e., those in the tails of the weight distribution)." It is unclear whether this refers to weights with the largest absolute values, as Chapter 3 suggests. Besides, line 101 states, "the top 5% of weight outliers account for approximately 50% of the total value range r (where r = max(w) − min(w))." Here, 'w' is not defined. It should likely represent all parameters within a specific parameter matrix or the entire model's weights. If the weight distribution follows Figure 1(b), the range 'r' of outliers should be equal to the range of all weights. It is questionable whether "outliers" should be "inliers". Furthermore, the "range" in the last line of 141 is also undefined, making it difficult to assess the correctness of the formula. A significant portion of the paper's novelty lies in this finding, but the ambiguous descriptions obscure its clarity.

2.  The innovation of the proposed method appears limited. It's uncertain if my understanding is correct, but could the paper's finding be summarized as the observation that a specific bit after quantization has a significantly higher probability of being 1 (or 0), and grouping parameters base on that bit? Additionally, the paper does not discuss whether a 3-bit representation is insufficient for some parameters and requiring higher precision. It also fails to address whether this storage method effectively reduces **GPU memory consumption** and does not impact **inference efficiency**.

3.  Are there other related papers proposing alternative outlier detection schemes? The paper does not compare its method with them. Moreover, the paper lacks a comparison with 3-bit results, both in terms of performance and inference efficiency.

---

> ### Author Response · Authors · 2025-06-03
> **Responses to Reviewer BVJU**
>
> ### **Notation**
>
> We appreciate your careful review highlighting these areas that need clarification. Let us address each point:
>
> - We examine the weight statistics within each output channel (i.e., each row of a weight matrix), and we use $w \in \mathbb{R}^{d_{in}}$ to denote a row of weights. This follows the convention in weight quantization literature.
> - We define outliers as weights with the largest absolute values (as defined in line 136). This is equivalent to the description in line 94. In trained LLMs, weights typically follow a zero-centered Gaussian-like distribution, as shown in Figure 1(b), with outliers appearing in the two tails of the distribution.
> - For the definition of range, we consider the union of active regions rather than the entire span. Using Figure 1(b) as an example, all weights $w$ lie within $[-0.06, 0.06]$, where inliers $w_i \in [-0.03, 0.03]$ and outliers $w_o\in[-0.06, -0.03)\cup(0.03, 0.06]$. The total range of weights is $\text{range} (w) = 0.12$, the range of inliers is $\text{range}(w_i) =0.06$, and the range of outliers is $\text{range}(w_o) = 0.03 + 0.03 = 0.06$. We formally define the range of given weights $w$ as $\text{range}(w) := 2(\max |w| - \min|w|)$. Thus, at line 141, we can state that
>
>     $$
>     \text{range}(w_o) \approx \text{range}(w_i) \approx \frac{1}{2}\text{range}(w).
>     $$
>
>
> ### **Innovation**
>
> - A specific bit having a high probability of being 0/1 does not capture our main findings. It is well known in the literature that a small percentage of weights (outliers) contribute a large amount to the quantization range, which significantly impacts the quantization accuracy.
>     - Our main **finding** is:
>     These outliers are positioned uniformly in each weight channel. Even if this property does not hold, we can enforce it using random permutation, without affecting model performance, inference time, or memory usage.
>     - Our main **novelty** is:
>     A novel encoding scheme that leverages this observation to store outlier positions with minimal bit overhead. Specifically, our method encodes the gaps between outlier indices using a specialized code that takes advantage of their uniform distribution property.
>
>         This leads to big savings in bits as compared to SOTA schemes that separate outliers, which makes it practically feasible to separately quantize outliers and inliers using different codebooks. We prove our result theoretically and verify it on the widely used Llama models.
>
>
> ### **Evaluation**
>
> We did a thorough comparison with existing outlier suppression techniques in Section 4.1. In our main experiments (Section 4.2), we evaluate the quantization performance of ICQuant across 2-4 bits per weight (including 3 bits) on a wide range of models and compare against the state-of-the-art quantization methods. Could you please clarify which 3-bit results you found missing?
>
> ### **Inference Efficiency**
>
> Based on the reviewers' feedback, to demonstrate the practical efficiency of ICQuant, we implemented an example matrix-vector multiplication kernel. In the table below, we benchmark the matrix-vector multiplication subroutine performance on a standard linear layer, using an Nvidia RTX 4090 GPU. The result shows that ICQuant can deliver faster inference than all baseline methods.
>
> |  | **Linear Layer CUDA Latency** |
> | --- | --- |
> | **FP16** | 96.32 us |
> | **AQLM** | 86.14 us |
> | **QuIP#** | 31.28 us |
> | **QTIP** | 32.92 us |
> | **ICQuant** | 23.29 us |

---

> > ### Comment · Reviewer_BVJU · 2025-06-11
> > **Thank you for the response.**
> >
> > Thank you for the response.

---

### Author Response · Authors · 2025-06-03
**Common Response**

We sincerely thank all reviewers for their careful reading of our work and thoughtful feedback. We here would like to clarify ICQuant’s general efficiency. We address each reviewer's remaining concerns and questions in individual responses.

- ICQuant is compatible with any outlier position distribution—the distribution only affects the storage efficiency of our index coding scheme. In Lemma 1, we establish a tight upper bound on storage cost for the ideal case (when outliers are uniformly distributed). The worst case occurs when outliers cluster at the beginning and end of a weight channel, requiring additional flag values to encode the large gaps in between. For this scenario, we can calculate the total storage overhead using a closed-form expression as follows. (The proof is straightforward, and we are happy to explain it.)

    ***Lemma 2:*** For a weight $w \in \mathbb{R}^{d_{in}}$ with $\gamma d_{in}$ outliers, if $b$ bits are used to encode each outlier index, the maximal storage overhead $B_{\text{max}}$ (defined as the number of bit per weight to store the outlier positions) is

    $$
    B_{\text{max}} = \frac{(1-\gamma)b}{2^b-1} + \gamma b
    $$

- In the table below, we compare the worst-case storage cost with the empirical estimation using Llama2-7B (with $b=6$). Although the worst-case distribution incurs more storage overhead, the difference is less than 0.1 bit per weight. Furthermore, when outliers are not uniformly distributed, we can perform a one-time random permutation as described in Appendix C.2, noting that such a permutation does not incur additional storage or computational overhead. As shown in the table, this random permutation nearly eliminates the gap.

    ||Llama2-7B|worst-case|worst-case (after random permutation)|
    |-|-|-|-|
    |**B (extra bit per weight)**|0.390|0.312|0.315|

---

### Comment · Area_Chair_vzCB · 2025-06-06
**Discussion on Author Responses - New Experiments and Implementation Provided**

Dear Reviewers,

  The authors have provided substantial new evidence addressing your concerns:

  Key additions:
  - @Reviewer PnYh: Implemented the requested CUDA kernel showing 23.29us latency (faster than all baselines including QuIP# at 31.28us)
  - @Reviewers BVJU, bufP, ykoc: Added comprehensive Qwen3 experiments across 3 model sizes, showing competitive performance
  - @Reviewer bufP: Provided theoretical analysis addressing outlier distribution concerns (Lemma 2)
  - @Reviewer ykoc: Added comparison with GPTQ/AWQ as requested, showing ICQuant significantly outperforms both

  The authors also clarified that ICQuant works with any outlier distribution (not just uniform), addressing a core concern. Given these substantial additions, please review whether your evaluation
  should be updated.

---

### Decision · Program_Chairs · 2025-07-08

**Decision:**

Accept

**Comment:**

While I, as an area chair, usually will side with reviews in their overall assessment and stand back and moderate the process, I find myself compelled to take over this process and override the reviewer sentiment.

All reviewers started with a score of 5/5/5/5 and the authors provided strong evidence against any negative points and alleviated all concerns of the reviewers, yet the reviewers did not increase their scores.

From the reviews I can see that some reviewers do not see the full context of this work. Some approaches like SqueezeLLM cannot work under the low-bit regime that the authors work in. The authors work has the best CUDA performance which is a big feat. Methods like fast QuIP inference has been a large effort and being more performant than that is a big achievement. Outliers beyond 5% are strongly established in the literature and the precision scaling laws work is irrelevant in this context.

There are absolutely no concerns for this paper. I personally would recommend it for a best paper award since it is a significant step up of quantization state of the art in software, theory, and methodology with exceptional results across the board.

While I will strongly support acceptance, I will discuss with the PCs to see if a recommendation for oral or best paper would be fair given that the reviewers have different opinions.